# The ubiquitin ligase Pellino1 targets STAT3 to regulate macrophage-mediated inflammation and tumor development

Soeun Hwang[1], Junhee Park[1], Seo-Young Koo[1], Si-Yeon Lee[1], Yunju Jo [2], Dongryeol Ryu [2], Heounjeong Go [3] & Chang-Woo Lee [1,4] ✉

Receptor-mediated signaling could be modulated by ubiquitination of pathway intermediates, but the role of such modification in the pathogenesis of inflammation and inflammation-related cancer is lesser known. The ubiquitin ligase Pellino1 has been shown to modulate immune signals by enabling various immune cells to respond to their receptor signals effectively. Here, we show that Pellino1 levels are elevated in patients with colitis, patients with colitis-associated colon cancer (CAC), and murine models of these conditions. In a monocyte-specific Pellino1 knock-out mouse model, we find reduced macrophage migration and activation, leading to attenuated development of colitis and CAC in male mice. Mechanistically, Pellino1 targets STAT3 for lysine 63-mediated ubiquitination, resulting in pathogenic activation of STAT3 signaling. Taken together, our findings reveal a macrophage-specific ubiquitination signaling axis in colitis and CAC development and suggest that Pellino1 is a potential candidate for treating chronic inflammation and inflammation-related cancer.

Ubiquitination has emerged as a significant mechanism for regulating various cellular functions within immune responses[1]. The fate of ubiquitination-mediated signaling components is manipulated through a variety of strategies, such as identifying protein degradation, initiating recycling processes, and regulating interactions with downstream effectors[2,3]. Although ubiquitination of proteins is crucial for downstream signal cascades initiated by receptor signaling, it is particularly significant in the context of signaling upon stimulation of membrane receptors of macrophages in innate immunity. However, related studies are insufficient.

Colitis-associated colorectal cancer (CAC) is a specific type of colorectal cancer (CRC) that develops from long-standing colitis in patients with inflammatory bowel disease (IBD)[4]. CAC causes 10–15% of deaths of patients with IBD, who have a 1.5–2.4-fold higher risk than the general population[5]. Intestinal macrophages have a role in maintaining tissue homeostasis during inflammation, especially in inducing resolution after inflammation[6]. However, these cells are also implicated in the pathogenesis of IBD[6]. In patients with IBD, CD68 macrophages can infiltrate the intestinal mucosa and spread extensively throughout thickened mucosa and submucosa[7]. These infiltrated macrophages secrete high levels of proinflammatory mediators, nitric oxide, reactive oxygen intermediates, and metalloproteases, leading to an intensive accumulation in the lamina propria[8]. Moreover, tumor-associated macrophages (TAM), which contribute to tumor growth, invasion, migration, and angiogenesis, are emerging as promising targets for CAC immunotherapy[9]. Therefore, a deeper understanding of molecular pathways involved in the differentiation and function of intestinal macrophages can help to identify a new class of targets to promote remission in patients with IBD and CAC.

[1]Department of Molecular Cell Biology, Samsung Medical Center, Sungkyunkwan University School of Medicine, Suwon 16419, South Korea. [2]Department of Biomedical Science and Engineering, Gwangju Institute of Science and Technology (GIST), Gwangju 61005, South Korea. [3]Department of Pathology, University of Ulsan College of Medicine, Asan Medical Center, Seoul 05505, South Korea. [4]Research Institute, Curogen Technology, Suwon 16419, South Korea. ✉e-mail: cwlee1234@skku.edu

Various inflammatory cytokines, including tumor necrosis factor (TNF) α, interferon (IFN) γ, and interleukins (IL-1, IL-6, and others), actively contribute to the progression of IBD[10]. Numerous cytokines act as ligands for cell surface receptors that activate cytoplasmic transcription factors, such as signal transducers and activators of transcription (STAT)[11]. In particular, STAT3 is a key factor in inflammatory signaling cascades in IBD[12]. Macrophage-specific STAT3 is directly relevant to intestinal inflammatory responses[13,14]. Activation of STAT3 in macrophages generally triggers the production of anti-inflammatory cytokines such as IL-10[15]. However, it can also stimulate the generation of proinflammatory cytokines such as IL-6, which exacerbate inflammation in IBD[16]. Thus, receptor-mediated signaling plays a vital role in the development of colitis and CAC. Nevertheless, the macrophage receptor-mediated signaling axis responsible for orchestrating chronic intestinal inflammation and cancer in response to various pathogens remains unclear.

Pellino1 is the most representative E3 ubiquitin ligase induced by various receptor signaling pathways[1,17,18]. Therefore, it is important to study ubiquitination control and cancer development mediated by receptor signaling through Pellino1 induction. Recent studies have unveiled a significant role of Pellino1 in activating pattern recognition receptors (PRR) such as toll-like receptors (TLR), T-cell receptors, and cytokine receptors to mediate inflammation and autoimmunity[17,19,20]. Moreover, abnormalities in Pellino1 activity are closely associated with a range of diseases, including B-cell lymphoma and lung cancer[21,22]. We initially found an aberrant upregulation of macrophage Pellino1 expression in inflammation, for example, inflamed intestinal tissues from both IBD patients and mice with DSS-induced colitis.

In this study, we establish monocyte-specific Pellino1-deficient mice to investigate the role of macrophage Pellino1 in the development of inflammation and inflammation-related cancer. Monocyte-specific Pellino1 ablation reduces macrophage infiltration, thereby inhibiting the development of colitis and CAC. Specifically, Pellino1 promotes macrophage migration by directly regulating the activation of STAT3 through lysine 63-mediated ubiquitination. Our findings reveal that macrophage Pellino1 plays an important role in macrophage function affecting the development of intestinal inflammation.

## Results

### Increased expression of macrophage Pellino1 during colitis development

To evaluate the aberrant expression of Pellino1 in inflammation, we first examined expression levels of Pellino1 in colonic mucosal samples obtained from patients with ulcerative colitis (UC) and Crohn's disease (CD). Elevated Pellino1 expression was observed in the epithelium of patients with both UC and CD (Supplementary Fig. 1a). Interestingly, the population of CD68+ cells expressing Pellino1 in the mucosa of the IBD patients was abundant, compared to the healthy control (Supplementary Fig. 1b, c). Pellino1 is predominantly expressed in CD68 macrophages in the mucosa of patients with IBD (Supplementary Fig. 1d). Consistently, Pellino1 expression in intestinal macrophages was increased in UC and CD colon samples compared to that in non-IBD colon samples based on immunofluorescence staining (Supplementary Fig. 1e).

As crucial protein receptors in innate immunity, TLRs can activate signaling pathways by recognizing pathogen-associated molecular patterns (PAMP) derived from microbes, thereby secreting inflammatory, anti-inflammatory cytokines or chemokines and regulating adaptive immunity[23]. To explore potential changes of Pellino1 expression in macrophages during TLR signaling, bone marrow-derived macrophages (BMDM) from Pellino1flox/flox mice (wild-type, WT) were treated with various TLR agonists. Results showed that the expression of Pellino1 was significantly elevated in BMDMs treated with various TLR agonists, particularly in BMDMs treated with lipopolysaccharide (LPS) (Fig. 1a, b). To determine how Pellino1 expression

actually changed during colitis, we developed a murine colitis model by administering dextran sodium sulfate (DSS) in the drinking water of mice. Intestinal tissues from colitis mice showed significant upregulation of Pellino1 levels compared to those from control mice (Fig. 1c, d). Immunofluorescence staining confirmed increases in Pellino1 expression in colonic F4/80 macrophages (Fig. 1e) and CD68 macrophages (Supplementary Fig. 2) after DSS challenge. Pellino1 mRNA expression was sharply elevated in intestinal macrophages from DSS-induced colitis mice compared to that from healthy mice (Fig. 1f), indicating that the expression level of Pellino1 in intestinal macrophages is closely associated with colonic inflammation.

### Deletion of macrophage Pellino1 attenuates experimental colitis

We generated myeloid-specific Pellino1 knock-out mice by crossing WT mice with transgenic mice expressing Cre under the control of a myeloid-specific promoter (LysM-Cre) (Fig. 2a–c and Supplementary Fig. 3a, b). Pellino1flox/flox; LysM-Cre (Pellino1-mKO) mice showed no obvious abnormalities in growth, survival, immune cell development, or populations compared to their littermate controls (Supplementary Fig. 3c–e). To investigate the role of macrophage Pellino1 in colitis, we induced acute colitis in WT and Pellino1-mKO mice with a 1.5% DSS challenge (Fig. 2d). In the assessment of disease activity index (DAI)[24], which combines scores based on the degree of diarrhea, rectal bleeding, and weight loss, Pellino1-mKO mice were less susceptible to DSS-induced colitis than WT mice (Fig. 2e, f). WT mice showed a loss of glandular structure in almost the entire colon, and the mucosa was replaced by a granuloma form, indicating the development of severe colitis. In contrast, Pellino1-mKO mice had localized accumulations of inflammatory cells and mucosal erosion in a few areas of the colon (Fig. 2g). Ki67 staining, a marker for cellular proliferation, revealed that Pellino1 deletion reduced DSS-induced cell proliferation in colon tissues (Fig. 2h). Notably, WT mice showed robust recruitment of F4/80 macrophages whereas Pellino1-mKO mice exhibited a great reduction in F4/80 macrophage infiltration after DSS treatment (Fig. 2h). However, WT and Pellino1-mKO mice showed no significant difference in the infiltration of Ly6G+ neutrophils (Fig. 2h). These data indicated that Pellino1-mKO mice exhibited significantly attenuated epithelial damage, macrophage infiltration, crypt distortion, and cell proliferation activity compared to WT mice. Consistent with reduced macrophage infiltration, levels of secretory pro-inflammatory cytokines TNFA, IL6, and IL1B were significantly decreased in colons of Pellino1-mKO mice compared to those in WT mice (Fig. 2i). Moreover, the expression of IL10, an anti-inflammatory cytokine known to play a protective role against tissue damage[25], was higher in Pellino1-mKO mice than in WT mice (Fig. 2j). In addition, spleens of Pellino1-mKO mice showed lower 16s ribosomal RNA (rRNA) bacterial load than those of WT mice, suggesting the preservation of epithelial barrier integrity in mice with myeloid-specific Pellino1 deficiency (Fig. 2k). Overall, these data imply that myeloid Pellino1 can exacerbate colitis and contribute to the accumulation of macrophages in response to inflammation.

### Down-regulation of macrophage Pellino1 improves prognosis and suppresses the development of CAC

To identify the association of colitis caused by Pellino1-mediated signaling in macrophages on CAC development, we generated a mouse model using azoxymethane (AOM)/DSS (Fig. 3a), a combination widely used in CAC research[26]. Mice were euthanized 12 weeks after AOM injection when tumors were clearly seen, as evidenced by the presence of adenocarcinoma with severe tumor-invasive leukocytes (Supplementary Fig. 4a, b). We further observed notable increases in expression levels of PCNA, β-catenin, and MDM2 widely recognized as cell proliferation markers associated with colon cancer in the AOM/DSS-treated mice (Supplementary Fig. 4c). These findings prompted us to evaluate the role of

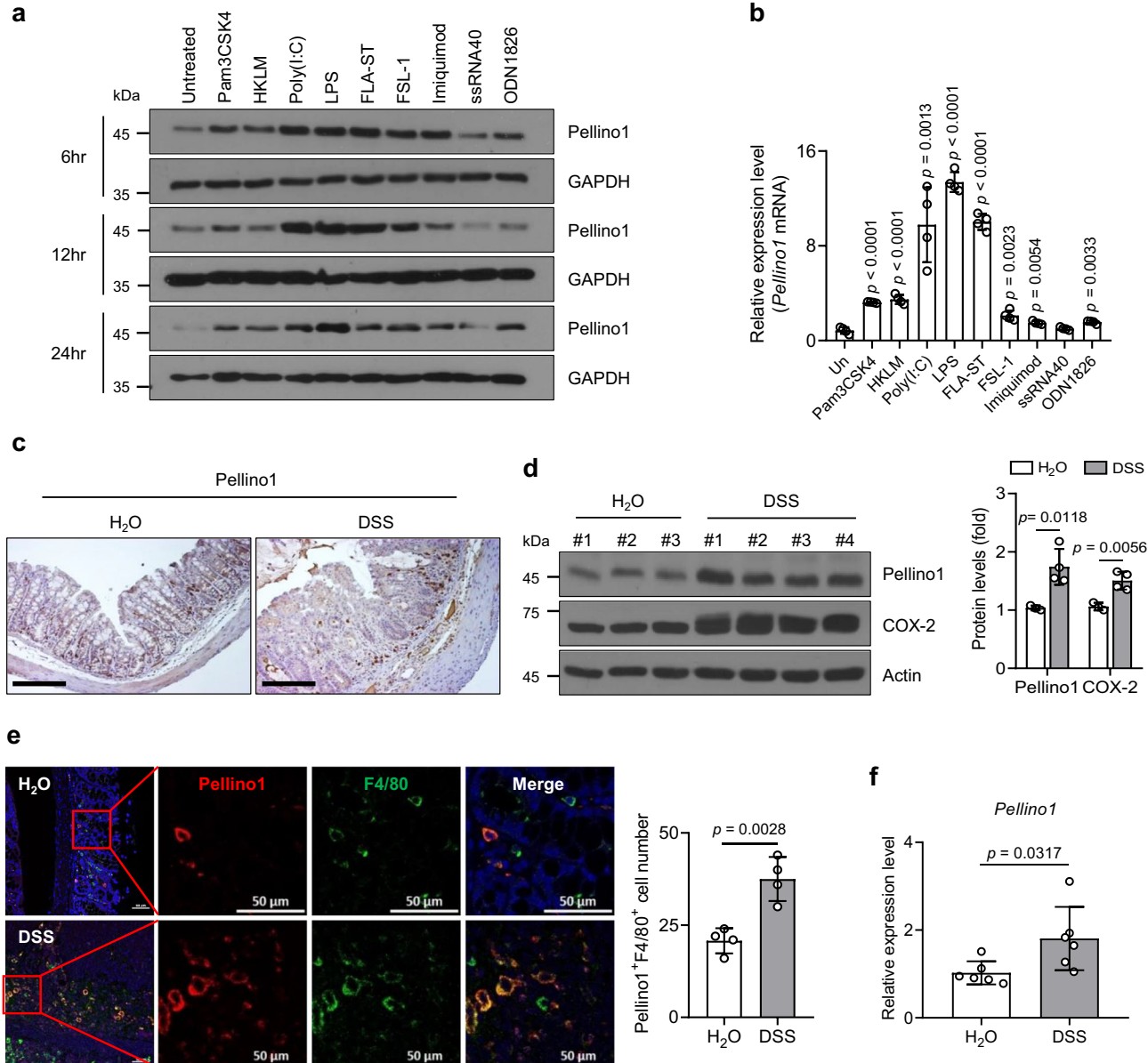

**Fig. 1 | Increased expression of macrophage Pellino1 during the development of colitis. a** Immunoblot analysis of Pellino1 protein levels in lysates of BMDMs from WT mice stimulated with TLR agonists (100 ng/mL Pam3CSK4, $10^7$ cells/mL HKLM, 1 µg/mL poly(I:C), 1 µg/mL LPS, 1 µg/mL ST-FLA, 50 ng/mL FSL-1, 2 µg/mL Imiquimod, 3 µg/mL ssRNA40, and 5 µM ODN1826) for the indicated times. TLR1/2, Pam3CSK4; TLR2, HKLM; TLR3, poly(I:C); TLR4, LPS; TLR5, ST-FLA; TLR2/6, FSL-1; TLR7, Imiquimod; TLR8, ssRNA40; and TLR9, ODN1826. **b** Expression levels of *Pellino1* mRNA in BMDMs from WT mice stimulated with TLR agonists for 6 h were measured and normalized to those of GAPDH ($n = 4$). **c** Representative immunohistochemical staining of Pellino1 in colon tissues obtained from WT mice drinking normal or 3% DSS water for 7 days. Scale bar = 200 µm. **d** (Left) Immunoblot analysis of Pellino1 and COX-2 protein levels in lysates extracted from intestines of WT male mice drinking normal or 1.5% DSS water for 9 days ($H_2O$, $n = 3$; DSS, $n = 4$). (Right) Quantification of Pellino1 and COX-2 protein levels. Pellino1 and COX-2 protein levels were normalized to Actin. **e** (Left) Immunofluorescence staining of Pellino1 (red), F4/80 (green), and DAPI (blue) in colon tissues of WT mice drinking normal or 1.5% DSS water for 9 days. DAPI stains the nuclei. Scale bar = 50 µm. (Right) Number of Pellino1⁺F4/80⁺ cells ($n = 4$). Quantification of cells in the field of view area. **f** Expression levels of *Pellino1* mRNA in intestinal macrophages obtained from WT male mice drinking normal or 1.5% DSS water for 9 days were measured and normalized to those of GAPDH ($n = 6$). Data were represented as mean ± SD in (**b**, **d**, **e**, **f**). All statistical comparisons were made using a two-tailed Student's *t*-test. Source data are provided as a Source Data file.

macrophage Pellino1 in the development of CAC. Interestingly, Pellino1-mKO mice exhibited a higher survival rate (Fig. 3b), less weight loss (Fig. 3c), and longer colons than WT mice (Fig. 3d and Supplementary Fig. 4a). Macroscopic images clearly showed reductions in tumor size and numbers in Pellino1-mKO mice compared to WT mice (Fig. 3e). We found that the total number and size of tumors in Pellino1-mKO mice were smaller than those in WT mice (Fig. 3f). Histological analysis additionally showed attenuated tubular adenoma, mucosal dysplasia, and reduced immune cell

infiltration in colon tissues of Pellino1-mKO mice (Fig. 3g and Supplementary Fig. 4b). We also observed a reduced expression of an inflammatory factor *IL1B* in colon tissues and lower *16s rRNA* bacterial loads in spleen of Pellino1-deficient mice (Fig. 3i, k). Additionally, Pellino1-mKO mice treated with AOM/DSS showed lower levels of *IL10* (Fig. 3j) known to promote immune evasion and contribute to cancer progression[27,28]. Taken together, these data suggested that Pellino1-mKO mice had reduced CAC development and improved anti-CAC microenvironment.

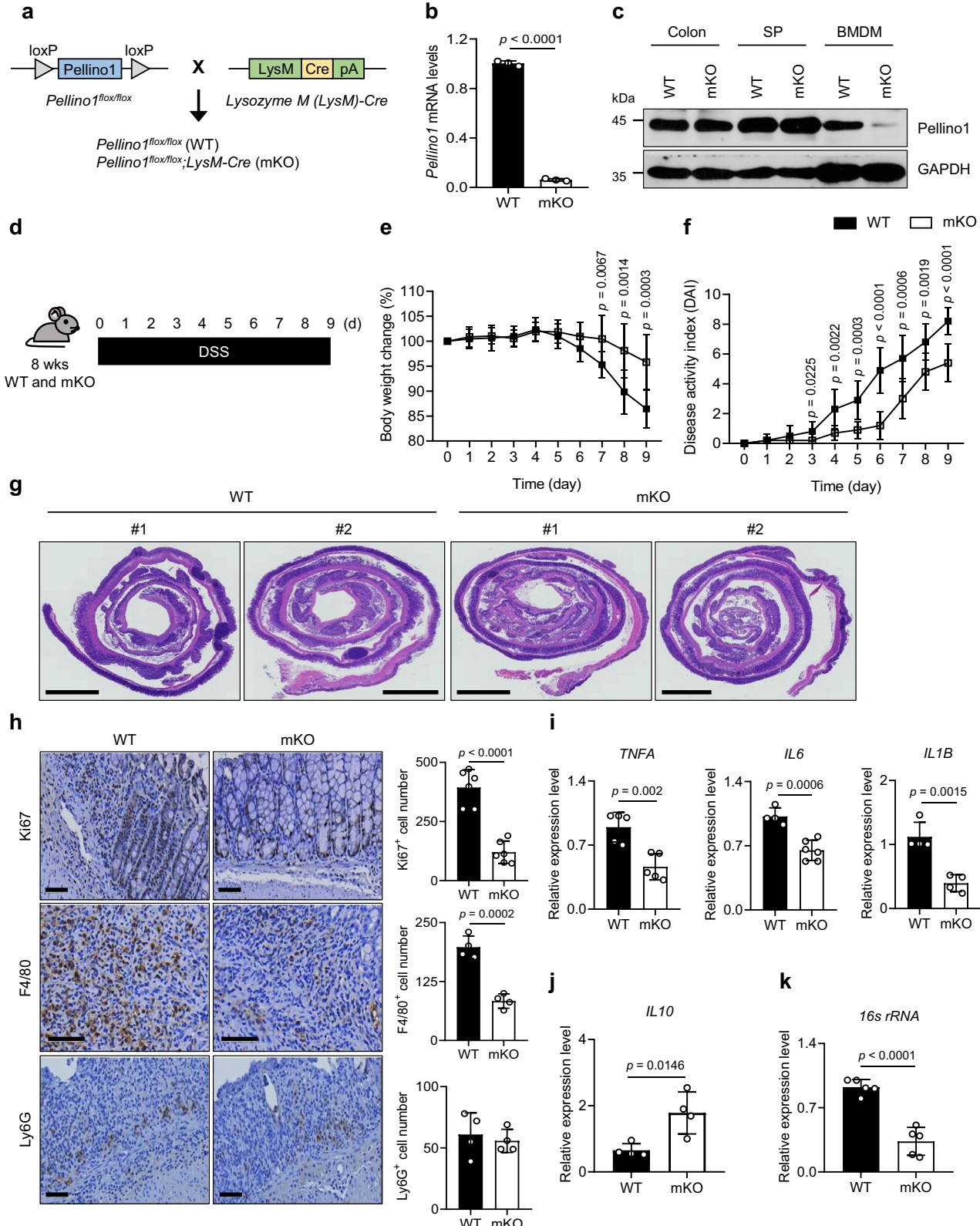

We hypothesized that these Pellino1-deficient mice phenotypes might be attributed to reduced macrophage accumulation in colonic mucosa. As expected, F4/80 macrophage infiltration into tumors was sharply decreased in Pellino1-mKO mice compared to that in WT mice (Fig. 3h). However, there was no significant difference in the number of Ly6G neutrophils in colonic tumor tissues between WT and Pellino1-mKO mice (Fig. 3h). We compared macrophage phenotypes between

WT and Pellino1-mKO mice to clarify whether the tumor-promoting activity of macrophages was decreased in Pellino1-mKO mice during CAC development. Given that TAMs are highly associated with M2 macrophage types[29], we speculated that Pellino1-mKO mice might have fewer M2 macrophages than WT mice. Indeed, M2-like phenotypes distinguished by CD206 expression were diminished in Pellino1-mKO mice compared to those in control mice (Supplementary Fig. 5a).

**Fig. 2 | Deletion of macrophage Pellino1 attenuates experimental colitis.**
**a** Schematic diagram of *Pellino1$^{flox/flox}$* (WT), *Lysozyme M* (LysM)-Cre, and *Pellino1$^{flox/flox}$; LysM-Cre* (Pellino1-mKO). **b** Expression levels of *Pellino1* mRNA in BMDMs from WT and Pellino1-mKO mice. *Pellino1* mRNA expression levels were assessed by qRT-PCR and normalized to GAPDH expression (*n* = 3). **c** Immunoblot analysis of Pellino1 protein levels in lysates from colons, spleens, and BMDMs of WT and Pellino1-mKO mice. GAPDH was used as a loading control. **d** Experimental schematics for the acute DSS-induced colitis model. WT and Pellino1-mKO male mice were administered 1.5% DSS for 9 days and euthanized on the 9th day. **e** Changes of body weights of WT and Pellino1-mKO male mice are expressed as percentages of their initial weights (*n* = 10). **f** Disease activity index (DAI) values for WT and Pellino1-mKO male mice were assessed daily as described in the "Methods" section (*n* = 10). **g** H&E-stained images of colon sections. Scale bar = 2 mm. **h** (Left) Representative immunohistochemical stained images of Ki67, F4/80, and Ly6G in colon tissues of WT

and Pellino1-mKO male mice on day 9 after DSS treatment. Scale bar = 60 µm. (Right) Numbers of Ki67 (*n* = 6), F4/80 (*n* = 4), and Ly6G (*n* = 4) positive cells. Quantification of cells was performed in the field of view area. **i** Levels of *TNFA* (*n* = 5), *IL6* (WT, *n* = 4; Pellino-mKO, *n* = 6), and *IL1B* (*n* = 4) mRNA were measured in colon tissues of WT and Pellino1-mKO male mice after DSS treatment. All mRNA expression levels were assessed by qRT-PCR and normalized to GAPDH expression. **j** *IL10* mRNA levels in colon tissues from WT and Pellino1-mKO male mice treated with DSS were analyzed using qRT-PCR and normalized to GAPDH expression (*n* = 5). **k** The mRNA expression level of *16s rRNA* in spleens of WT and Pellino1-mKO male mice after DSS treatment. The mRNA expression was assessed by qRT-PCR and normalized to GAPDH expression (*n* = 5). Data were represented as mean ± SD in (**b**, **e**, **f**, **h**, **i**, **j**, **k**). All statistical comparisons were made using a two-tailed Student's *t*-test. Source data are provided as a Source Data file.

However, the M1-like phenotype population, characterized by iNOS expression, was similar between WT and Pellino1-mKO mice (Supplementary Fig. 5a). A series of additional M2 marker analyses confirmed a significant decrease in the M2-like phenotype in Pellino1-deficient colon tissues, whereas M1 marker analysis showed no significant difference between WT and Pellino1-deficient colon tissues (Supplementary Fig. 5b, c). To further investigate whether Pellino1 directly impacts macrophage polarization, we treated WT and Pellino1-deficient macrophages with specific cytokines: LPS and IFN γ to induce M1 macrophages, and IL-4 and IL-13 to induce M2 macrophages. We then evaluated the expression levels of M1 and M2 marker genes. As shown in Supplementary Fig. 6a, there was no significant difference in the expression of M1 markers (*NOS2*, *CD86*, *TNFA*) between WT and Pellino1-mKO mice. However, gene expression levels of M2 markers (*ARG1*, *CD163*, *IL10*) were significantly lower in Pellino1-mKO mice than in WT mice (Supplementary Fig. 6b). These results indicate that myeloid-specific Pellino1 can promote CAC development by enhancing the infiltration and tumor-promoting activity of M2 macrophages.

### Pellino1 depends on inflammation to induce CAC

To investigate whether the development of Pellino1-mediated CAC was a prerequisite for inflammation, we next established a colitis-independent model treated only with AOM (Supplementary Fig. 7a). However, little differences in body weight change, colon length, or the number of tumor nodules were seen between WT and Pellino1-mKO AOM-induced mice (Supplementary Fig. 7b–d). Further biochemical analyses confirmed no significant differences between AOM-induced colon cancer-bearing WT and Pellino1-mKO mice (Supplementary Fig. 7e, f). The population of F4/80 macrophages was also similar between WT and Pellino1-mKO mice treated with AOM (Supplementary Fig. 7g). Next, we administered a low concentration of DSS to WT and Pellino1-mKO mice to induce chronic colitis (Supplementary Fig. 8a). Although there were no differences in body weight change or colon length between WT and Pellino1-mKO mice (Supplementary Fig. 8b, c), fewer colon polyps were observed in Pellino1-deficient mice (Supplementary Fig. 8d). Histological examination using H&E staining revealed smaller regions of tissue destruction with altered crypt structures accompanied by reduced immune cell infiltration in colonic tissues of Pellino1-mKO mice (Supplementary Fig. 8f). Consistently, expression levels of inflammatory markers were diminished in Pellino1-mKO mice (Supplementary Fig. 8e). We also found that Pellino1 deficiency reduced macrophage recruitment in chronically inflamed tissues (Supplementary Fig. 8g). Taken together, these findings indicate that activation and infiltration of macrophages induced by Pellino1 can promote inflammatory responses, subsequently impacting inflammation-related cancer.

### Pellino1 expression is associated with poor CRC prognosis

Considering that aberrant upregulation of Pellino1 is closely associated with B-cell lymphoma[21], lung cancer[22], and breast cancer[30], we assumed

that Pellino1 expression might also be increased in colon cancer. As shown in Fig. 4a, Pellino1 was drastically increased in AOM/DSS-induced CAC, exceeding levels observed in colitis. Thus, we evaluated *Pellino1* expression in human colorectal adenocarcinoma using Kaplan–Meier survival analysis to explore the prognostic significance of Pellino1 in CRC patients. *Pellino1*-high patients had markedly reduced survival probability compared to *Pellino1*-low patients (*p* = 0.0019; Fig. 4b). Specifically, among male patients, *Pellino1*-high expression was correlated with a substantial decline in the overall survival rate (OS), resulting in an 8-year survival rate of 0%, in contrast to 61% for the *Pellino1*-low expression group (*p* = 0.046, Fig. 4c). While the correlation between increased Pellino1 expression and decreased patient survival rates was insufficient, particularly in stage I, *Pellino1*-high expression was directly linked to reduced patient survival rates, particularly in stages III and IV (Fig. 4d, e). Overall, these results suggest direct correlations of Pellino1 expression with prognosis and cancer staging in patients with colon adenocarcinomas.

### Pellino1 deficiency inhibits the migration of macrophages during colitis and CAC

Since extensive macrophage infiltration is associated with disease progression and poor prognosis[31], we focused on reduced recruitment of macrophages in Pellino1-mKO mice with colitis and CAC. Thus, we analyzed CD11b$^+$CX3CR1$^{intermediate (int)}$ migratory macrophages and CD11b$^+$CX3CR1$^{high (hi)}$ resident macrophages to examine distinct subsets of intestinal macrophages present under various conditions (Supplementary Fig. 9). The accumulation of CD11b$^+$CX3CR1$^{int}$ migratory macrophages increased in WT mice with the severity of intestinal inflammation (Fig. 5a). Notably, Pellino1-mKO mice displayed reduced levels of CD11b$^+$CX3CR1$^{int}$ cells compared to their WT littermates. However, there was no difference in the frequency of CD11b$^+$CX3CR1$^{hi}$ resident macrophages between WT and Pellino1-mKO mice under any conditions (Fig. 5b). Considering that Pellino1 could affect macrophage migration, we hypothesized that Pellino1-deficient macrophages might have impaired recognition of danger signals that could hinder migration toward inflammatory sites. Macrophages are known to express PRRs such as TLR2, TLR4, CD36, and Dectin-1, which enable them to detect external ligands during the early stages of immune responses[32]. To explore this possibility, we investigated whether Pellino1 controls macrophage migration to inflamed sites by influencing the expression of PRRs (Supplementary Fig. 10a, b). Our results showed no significant difference in PRR expression between WT and Pellino1-mKO mice under normal or CAC state. To validate the effect of Pellino1 on macrophage migration, we performed in vitro wound healing and migration assays to examine the motility of BMDMs in response to chemical gradients of immune-related signals. While the WT group exhibited a substantial, time-dependent increase in migration in response to treatment with LPS, CCL5, or transforming growth factor (TGF) β, the Pellino1-mKO group demonstrated a significantly attenuated migratory capacity (Fig. 5c and Supplementary Fig. 11).

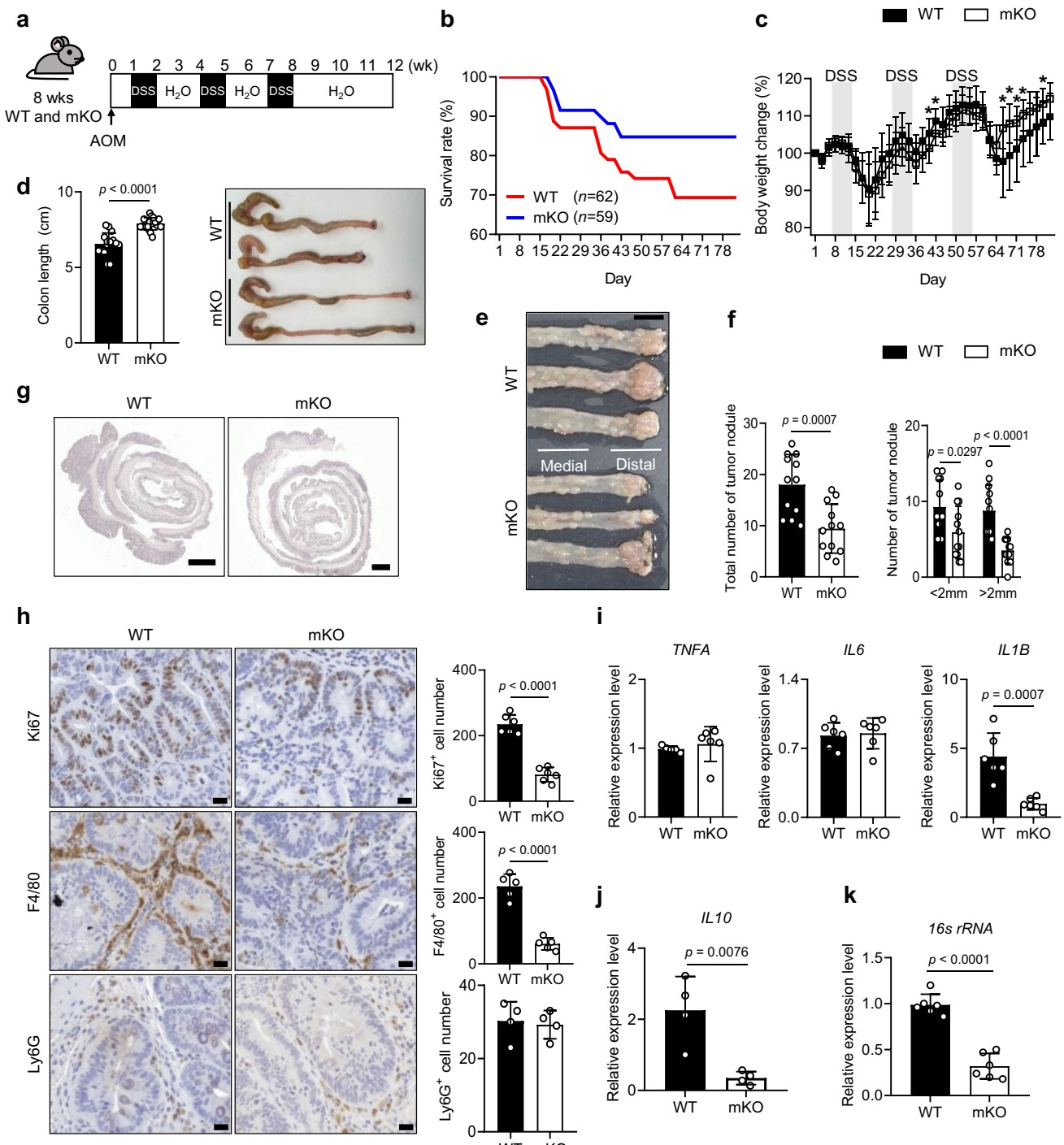

**Fig. 3 | Down-regulation of macrophage Pellino1 improves prognosis and suppresses the development of CAC. a** Experimental schematics for the AOM/DSS-induced CAC model. **b** Overall survival curves of WT and Pellino1-mKO male mice in the AOM/DSS model (WT, $n = 62$; Pellino-mKO, $n = 59$). **c** Body weights of AOM/DSS-induced CAC mice were monitored (WT, $n = 9$; Pellino-mKO, $n = 11$). *$p < 0.05$. Exact $p$ values are as follows: Day 40 = 0.0464, Day 43 = 0.034, Day 66 = 0.0189, Day 68 = 0.017, Day 71 = 0.0338, Day 73 = 0.0328, and Day 80 = 0.0284. **d** (Left) Colon lengths of WT and Pellino1-mKO male mice after AOM/DSS treatment ($n = 17$). (Right) Images of colon tissues from WT and Pellino1-mKO male mice after AOM/DSS treatment. **e** Representative images of colon tumor formation in WT and Pellino1-mKO male mice. These images indicate the medial and distal regions of the colon. Scale bar = 1 cm. **f** (Left) Total number of tumor nodules in the entire colon ($n = 12$). (Right) Numbers of small tumors (<2 mm) and large tumors (>2 mm) in the entire colon ($n = 12$). **g** H&E-stained images of colon sections from WT and Pellino1-mKO male mice. Scale bar = 400 μm. **h** (Left) Representative immunohistochemical stained images of Ki67, F4/80, and Ly6G in colon tissues of WT and Pellino1-mKO male mice after AOM/DSS treatment. Scale bar = 60 μm. (Right) Numbers of Ki67 ($n = 6$), F4/80 ($n = 5$), and Ly6G ($n = 4$) positive cells. Quantification of cells in the field of view area. **i, j** Levels of *TNFA* (WT, $n = 5$; Pellino-mKO, $n = 6$), *IL6* ($n = 6$), *IL1B* ($n = 6$), and *IL10* ($n = 4$) mRNA were measured in colon tissues of WT and Pellino1-mKO male mice after AOM/DSS treatment using qRT-PCR and normalized to GAPDH expression. **k** The mRNA expression level of *16s rRNA* in spleens of WT and Pellino1-mKO male mice after AOM/DSS treatment. The mRNA expression was assessed by qRT-PCR and normalized to GAPDH expression ($n = 6$). Data were represented as mean ± SD in (**c, d, f, h, i, j,** and **k**). All statistical comparisons were made using a two-tailed Student's $t$-test. Source data are provided as a Source Data file.

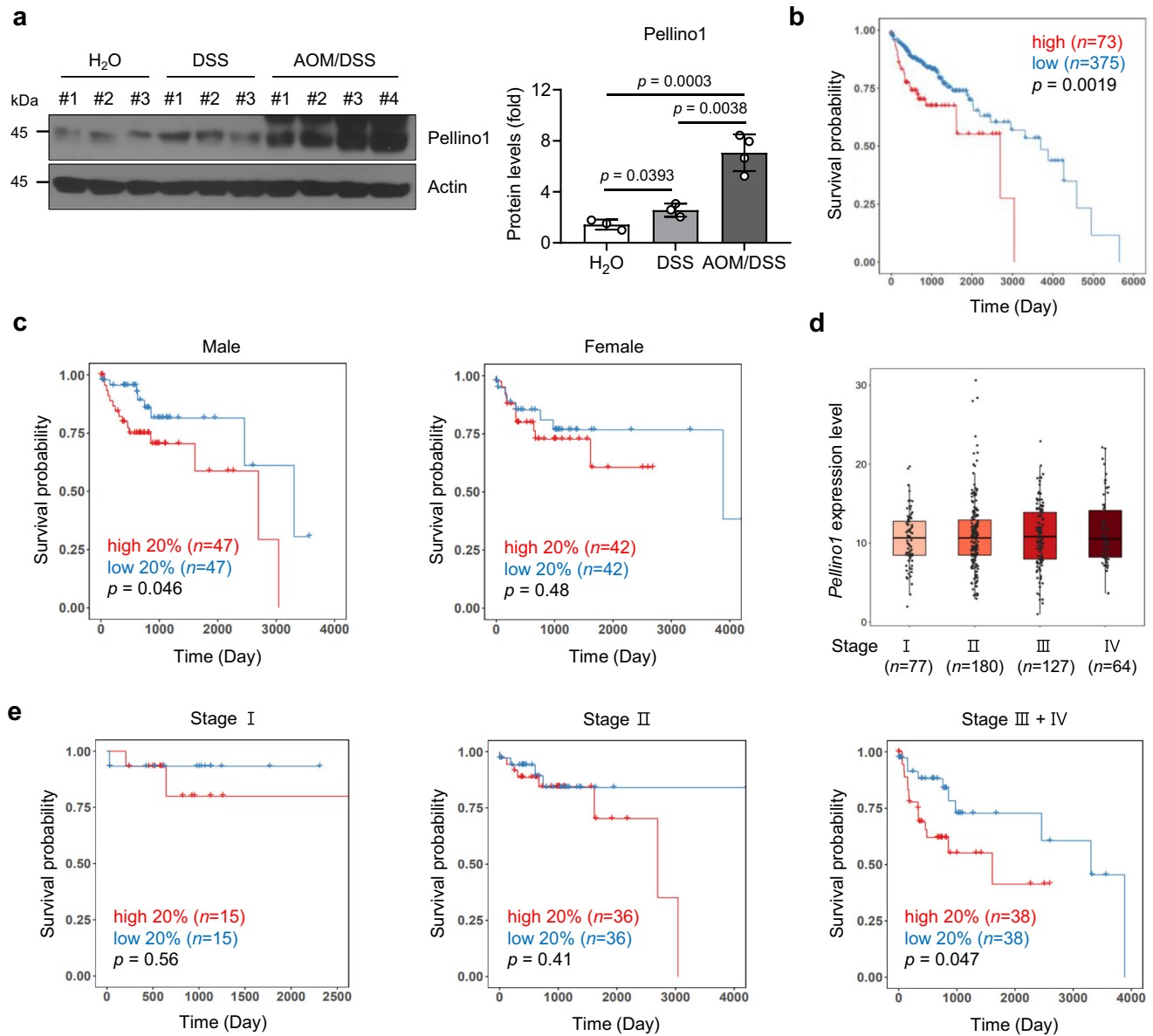

**Fig. 4 | Pellino1 expression is associated with poor CRC prognosis. a** (Left) Immunoblot analysis of Pellino1 protein levels in lysates obtained from colon tissues of WT male mice in normal, acute 1.5% DSS, and AOM/DSS groups. Actin was used as a loading control. (Right) Quantification of Pellino1 protein levels normalized to Actin levels (H$_2$O, $n = 3$; DSS, $n = 3$; AOM/DSS, $n = 4$). Data are presented as mean ± SD. Statistical comparisons were made using two-tailed Student's $t$-test and one-way ANOVA. **b, c** Kaplan–Meier survival analysis of patients with high or low *Pellino1* gene expression in colorectal adenocarcinoma from The Cancer Genome Atlas (TCGA-COAD) dataset. **b** Survival curves of groups divided by COAD overall optimal survival cutoffs and (**c**) survival curves of the top (high) and bottom (low) 20% groups based on *Pellino1* expression. **d** *Pellino1* expression levels at COAD stage. The boxplots indicate minimum and maximum (whiskers), 25th and 75th percentiles (bounds of box), and median (center). **e** Kaplan–Meier survival analysis of patients with high (top 20%) or low (bottom 20%) *Pellino1* gene expression at the indicated COAD stage. Kaplan–Meier survival curves (**b, c, e**) were analyzed using the two-sided log-rank test for statistical significance. Source data are provided as a Source Data file.

Statistical analysis revealed a significant decrease in the number of migrating macrophages in Pellino1-mKO BMDMs compared to that in WT BMDMs after treatment with LPS, CCL5, or TGF β (Fig. 5d). These data imply that Pellino1 plays an essential role in regulating macrophage migration.

We further investigated the effect of Pellino1 on phagocytosis, a primary function of resident macrophages. Phagocytosis is a pivotal process in the immune system for the effective clearance of pathogens, cellular debris, and other external particles. It is primarily orchestrated by macrophages, especially resident macrophages[33]. Although there was no significant difference in the frequency of resident macrophages between WT and Pellino1-mKO mice (Fig. 5a, b), expression levels of phagocytosis-related genes (*MFGE8*, *TIMD4*, and *ANXA1*) were increased in Pellino1-deficient macrophages compared to those in their WT counterparts under AOM/DSS conditions (Supplementary Fig. 12a). Consistent with higher mRNA expression levels of phagocytosis-related genes, expression levels of these genes were notably elevated in Pellino1-deficient BMDMs treated with LPS compared to those in WT BMDMs (Supplementary Fig. 12b). We cocultured CFSE-labeled MC38 cells with BMDMs to elucidate the influence of Pellino1 on the phagocytic capacity of macrophages. Pellino1-deficient BMDMs exhibited significantly higher phagocytic activity than WT BMDMs under LPS stimulation (Fig. 5e). Given that Pellino1 deficiency in macrophages enhanced the efficiency of antigen clearance, we speculated that Pellino1-mKO mice might produce fewer cellular factors required for macrophage migration. The recruitment

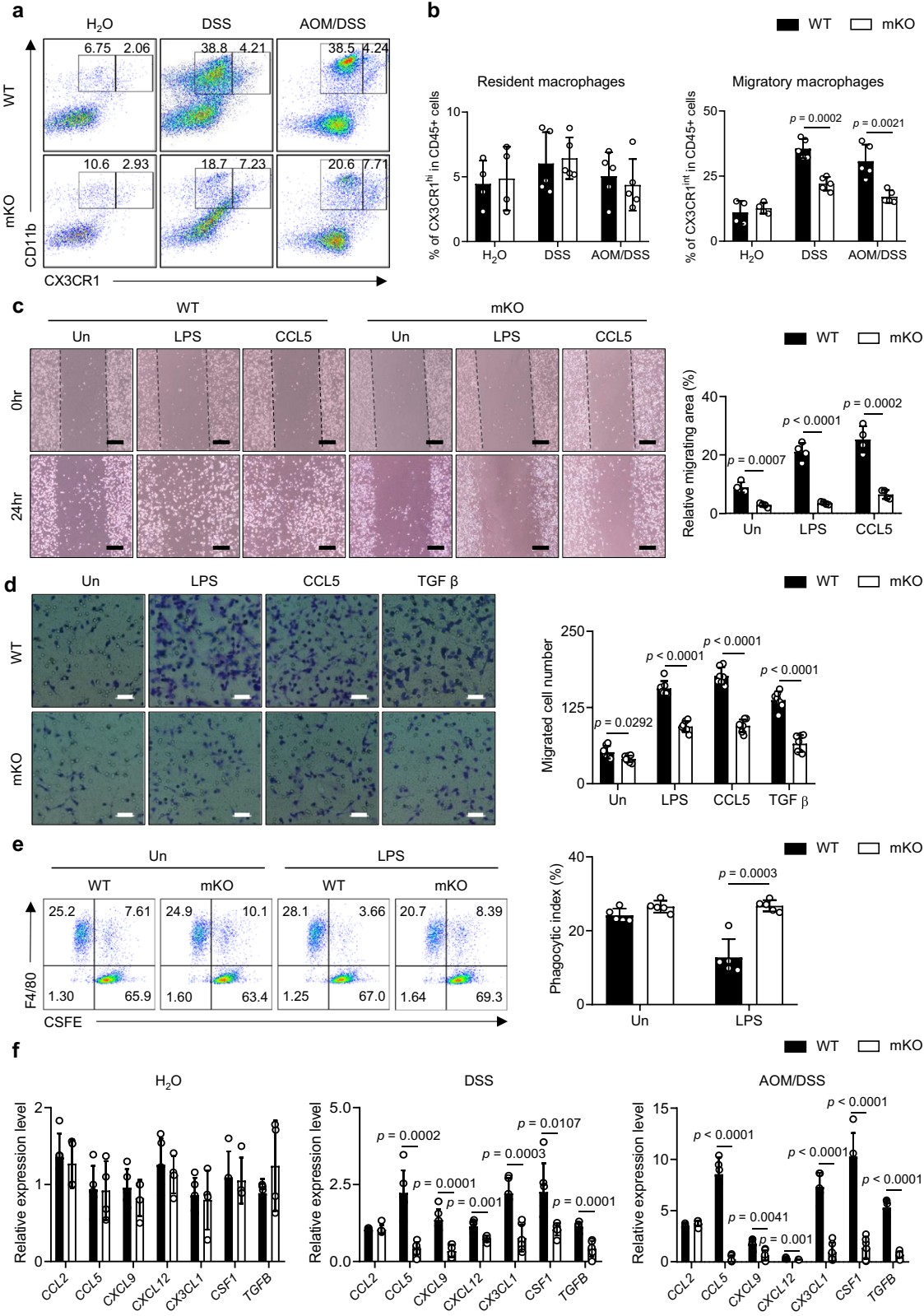

of monocytes/macrophages to damaged regions involves various factors, including cytokines, chemokines, and growth factors. In particular, CCL2, CCL5, CX3CL1, and macrophage colony-stimulating factor (M-CSF) are known to be essential for macrophage migration[34]. Importantly, during inflammation, mRNA expression levels of cytokines and chemokines associated with macrophage migration, such as

CCL5, CXCL9, CXCL12, CX3CL1, CSF1, and TGFB, were lower in intestines of Pellino1-deficient mice than in those of WT mice (Fig. 5f). Enhanced interactions between surface receptors on macrophages and cytokines/chemokines could amplify inflammatory signaling pathways, resulting in the recruitment, activation, and prolonged inflammation of immune cells[35]. Thus, we isolated colon macrophages from WT and

**Fig. 5 | Pellino1 deficiency inhibits the migration of macrophages during colitis and CAC. a** Representative flow cytometry plots gated on CD45$^+$ living cells isolated from colonic lamina propria of WT and Pellino1-mKO male mice in normal, acute 1.5% DSS, and AOM/DSS groups. Migratory macrophages (CD11b$^+$CX3CR1$^{int}$), resident macrophages (CD11b$^+$CX3CR1$^{hi}$). **b** (Left) Percentage of CD11b$^+$CX3CR1$^{hi}$ macrophages, (Right) CD11b$^+$CX3CR1$^{int}$ macrophage infiltration into the colonic lamina propria in WT and Pellino1-mKO male mice in normal, acute 1.5% DSS, and AOM/DSS groups (H$_2$O, $n = 4$; DSS, $n = 5$; AOM/DSS, $n = 5$). These percentages for both CD11b$^+$CX3CR1$^{hi}$ and CD11b$^+$CX3CR1$^{int}$ were determined by gating among CD45$^+$ cells. **c** (Left) Representative images of the wound healing assay. Wound healing assays were performed to examine the migration of BMDMs of WT and Pellino1-mKO under stimulation of 100 ng/mL LPS and 20 ng/mL CCL5 for 24 h. Initial wounded areas were marked with a dashed line. Scale bar = 100 μm. (Right) Quantification of migration areas using ImageJ ($n = 4$). **d** (Left) Representative images of the migration assay. BMDMs from WT and Pellino1-mKO mice were induced to migrate with stimulation from 100 ng/mL LPS, 20 ng/mL CCL5, and 100 ng/mL TGF β for 24 h. Scale bar = 50 μm. (Right) Quantification of migrated cells within the field of view area using ImageJ ($n = 7$). **e** (Left) CFSE-labeled MC38 tumor cells were co-cultured with BMDMs from WT and Pellino1-mKO mice for 6 h in the presence of 100 ng/mL LPS. (Right) Phagocytic index was calculated using the following formula: phagocytic index (%) = (number of F4/80$^+$CSFE$^+$ cells)/ (number of F4/80$^+$ cells) × 100 ($n = 5$). **f** Expression levels of mRNA associated with macrophage migration in colon tissues of WT and Pellino1-mKO male mice of normal (WT, $n = 5$; Pellino-mKO, $n = 4$), acute 1.5% DSS ($n = 6$), and AOM/DSS (WT, $n = 5$; Pellino-mKO, $n = 6$) groups were assessed by qRT-PCR and normalized to GAPDH expression. Data were represented as mean ± SD in (**b**–**f**). All statistical comparisons were made using two-tailed Student's $t$-test. Source data are provided as a Source Data file.

Pellino1-mKO mice and examined levels of several protein kinases (Supplementary Fig. 13). Expression levels of most protein kinases, including p-JNK, p-p38, p-AKT (S473), and p-AKT (T308), were decreased in Pellino1-mKO mice with CAC, while those of p-ERK and p-AKT (T308) were decreased in the same mice with colitis (Supplementary Fig. 13). Collectively, these results underscore the indispensable role of oncogenic macrophage Pellino1 in macrophage migration.

## Pellino1 functionally interacts with STAT3 signaling

In our initial investigation into molecular mechanisms underlying Pellino1 function, we examined expression profiles of representative marker molecules involved in signaling pathways in colitis and CAC using intestinal macrophages isolated from WT and Pellino1-mKO mice (Supplementary Fig. 13). Surprisingly, WT and Pellino1-mKO macrophages showed a major difference in the induction of STAT3 (Fig. 6a), a key regulator in the development of inflammation and inflammation-related cancer[36–38]. Pellino1-deficient macrophages showed a significant reduction in activated STAT3 (p-STAT3 Y705), particularly in the CAC group (Fig. 6a). Furthermore, we observed a strong positive correlation between p-STAT3 Y705 and Pellino1 expression in LPS-treated BMDMs, with WT BMDMs showing higher p-STAT3 Y705 levels than Pellino1-mKO BMDMs (Fig. 6b). Elevated levels of p-STAT3 at Y705 have been prominently associated with various types of cancer[39–41]. Recent studies suggest that NOD2 can influence STAT3 activation in IBD and colon cancer[42], leading us to investigate whether Pellino1 could affect p-STAT3 via NOD2 (Supplementary Fig. 14a, b). However, there was no significant difference in NOD2 expression between WT and Pellino1-deficient macrophages, and treatment with an NOD2 inhibitor also did not affect Pellino1 expression (Supplementary Fig. 14c, d). These results suggest that Pellino1 is unlikely to regulate STAT3 activation via NOD2.

To further investigate whether Pellino1-mediated macrophage migration depends on STAT3 signaling, we utilized S3I-201, an inhibitor of STAT3, to assess the migratory capability of WT and Pellino1-deficient macrophages (Supplementary Fig. 15a, b). S3I-201 did not affect Pellino1 expression in WT BMDMs after 1 h of LPS stimulation, although it slightly decreased LPS-induced expression of Pellino1 after 3 h (Supplementary Fig. 15a). Additionally, S3I-201 inhibited the migratory capacity of macrophages under stimulation with LPS, CCL5, or TGF β (Supplementary Fig. 15c, d). In Pellino1 deficiency, treatment with S3I-201 also suppressed macrophage movement under the same stimuli (Supplemented Fig. 15c, d). These findings suggest that the Pellino1-STAT3 signaling axis serves as a key regulator of macrophage migration.

To elucidate the mechanism underlying the regulation of STAT3 by Pellino1, we investigated the possibility of a direct protein–protein interaction between Pellino1 and STAT3. An in vitro binding assay using cellular extracts from BMDMs and RAW macrophage cells incubated with glutathione S-transferase (GST) or GST-STAT3 fusion

proteins revealed a direct interaction between the two proteins (Fig. 6c and Supplementary Fig. 16a). Co-immunoprecipitation assays confirmed the interaction between Pellino1 and STAT3 (Fig. 6d). We also found a direct binding between purified Pellino1 and STAT3 (Fig. 6e). Subsequent immunofluorescence assay using LPS-stimulated WT BMDMs showed a co-localization of p-STAT3 and Pellino1 (Fig. 6f and Supplementary Fig. 16b). Given that Pellino1 preferentially interacts with phosphorylated proteins[43], we hypothesized that Pellino1 might exhibit a stronger affinity for hyperphosphorylated STAT3 (p-STAT3) than for hypophosphorylated STAT3 (STAT3). Consequently, a GST pull-down assay was performed using RAW cells. Results revealed that GST-Pellino1 exhibited a higher binding affinity with LPS-stimulated Flag-STAT3 than with unstimulated Flag-STAT3 (Fig. 6g). Similarly, in vivo interaction between Pellino1 and p-STAT3 was further enhanced in RAW cells following an LPS challenge (Fig. 6h). Considering that phosphorylation on Tyr705 is a key regulatory site for STAT3 activation in response to various extracellular stimuli[44], we investigated a potential interaction of Pellino1 with p-STAT3 Y705. HEK293T cells were transfected with an expression plasmid encoding either WT STAT3 (WT-STAT3) or a dominant-negative variant of p-STAT3 Y705F (DN-STAT3). As shown in Fig. 6i, Pellino1 formed a complex more readily with WT-STAT3 than with DN-STAT3, suggesting that the mutation in STAT3 can induce structural or functional impairments that prevent Pellino1 from recognizing the binding site or reducing the binding affinity. Taken together, these results support a direct functional interaction between Pellino1 and STAT3, thereby augmenting the pathogenic activation of STAT3 and consequently promoting the migratory behavior of cancerous macrophages.

## Pellino1 mediates lysine 63-linked ubiquitination of STAT3 to enhance its activity

To investigate whether Pellino1 functions as an E3 ubiquitin ligase for STAT3, we transfected HEK293T cells with expression plasmids encoding Myc-tagged full-length Pellino1 (Myc-Pellino1 FL) or a Pellino1 C-terminal RING domain deletion mutant (Myc-Pellino1 ΔC), together with Flag-tagged STAT3 (Flag-STAT3) and HA-ubiquitin (HA-Ub) expression plasmids (Fig. 7a). Overexpression of Pellino1 FL clearly induced STAT3 ubiquitination, whereas overexpression of Pellino1 ΔC did not, indicating a functional role of Pellino1 as an E3 ubiquitin ligase for STAT3. We also observed that purified Pellino1 directly catalyzed STAT3 ubiquitination in vitro (Fig. 7b). Additionally, ubiquitinated STAT3 was detected in both cytoplasmic and nuclear fractions, showing higher levels in the nuclei of LPS-treated cells (Supplementary Fig. 16c).

Seven lysine residues (K6, K11, K27, K29, K33, K48, and K63) of ubiquitin are utilized to modify proteins through various linkage types[45]. Modification of lysine 48 (K48) and lysine 63 (K63) linkages is a widely recognized mechanism for regulating protein degradation and signal transduction, respectively[46,47]. To determine the specific lysine linkage type responsible for Pellino1-mediated STAT3 ubiquitination,

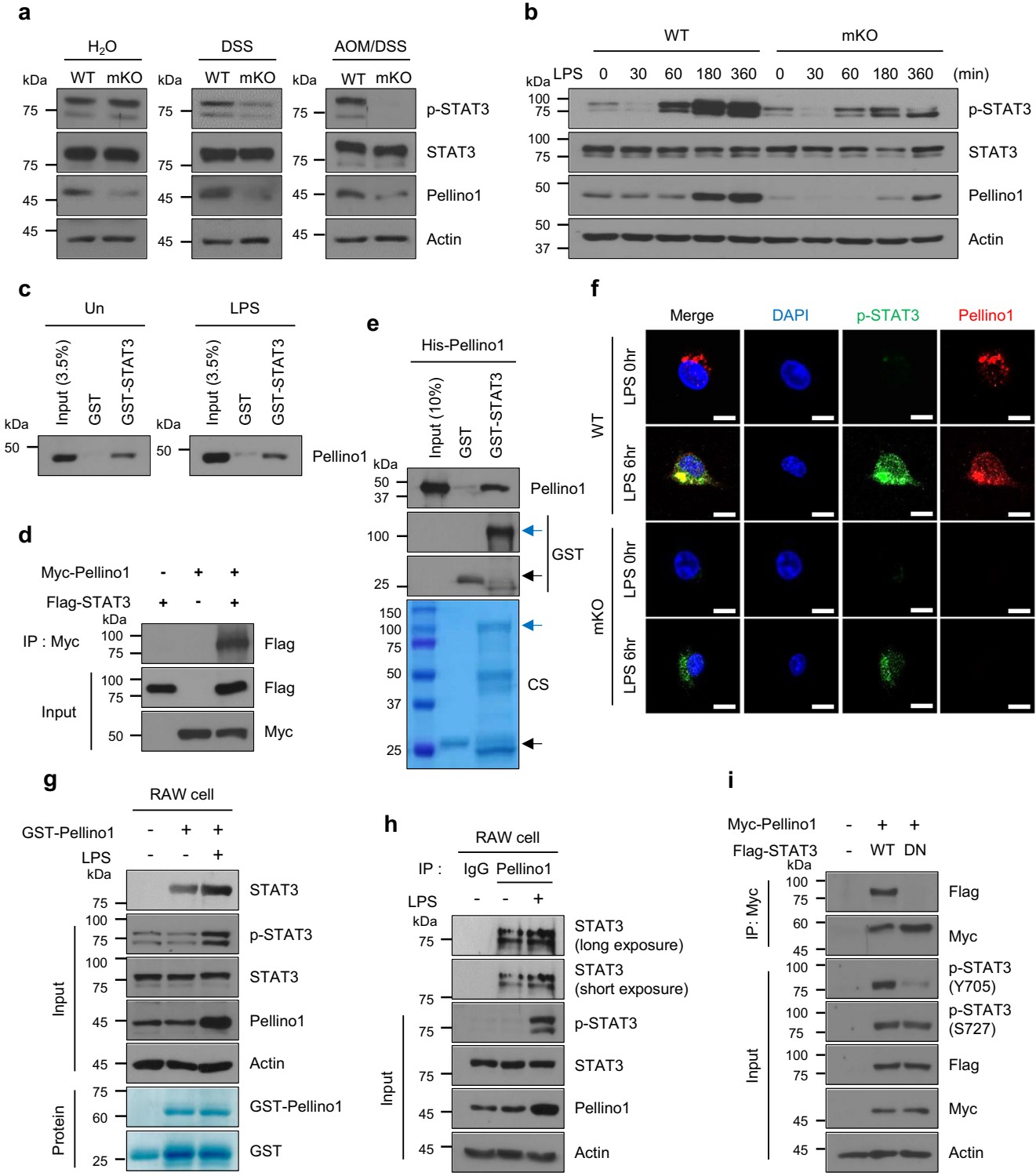

we co-transfected HEK293T cells with expression plasmids encoding HA-Ub WT, K48, and K63 (containing only K48 or K63, respectively) along with Myc-Pellino1 and Flag-STAT3 expression plasmids (Fig. 7c). Clearly, Pellino1 promoted ubiquitination of STAT3 in the presence of HA-Ub K63. Lysine-to-arginine ubiquitin mutants are ideal for studying biological processes involving a particular ubiquitin chain linkage[48]. We consistently observed that Pellino1 was associated with K63-linked ubiquitination of STAT3, as demonstrated using HA-Ub WT, mutant K48 Ub (K48 replaced by R; K48R), and mutant K63 Ub (K63 replaced by R; K63R) (Fig. 7d). Additionally, Pellino1-deficient BMDMs exhibited a markedly reduced capability to induce intracellular K63-linked

polyubiquitination of endogenous STAT3 compared to WT BMDMs (Supplementary Fig. 16d). Consistent with these findings, ubiquitination of STAT3 was clearly observed in WT macrophages. However, it was sharply reduced in Pellino1-mKO macrophages compared to that in WT macrophages (Fig. 7e). These results demonstrate that Pellino1 directly regulates K63-linked ubiquitination of STAT3.

Given that Pellino1 preferentially recognized p-STAT3 (Fig. 6g, h) and that p-STAT3 levels were reduced in Pellino1-mKO compared to those in WT macrophages (Fig. 6a, b), we further compared ubiquitination levels of p-STAT3 between untreated and LPS-treated WT and Pellino1-deficient BMDMs (Fig. 7f). As expected, ubiquitination levels

**Fig. 6 | Pellino1 functionally interacts with STAT3 signaling. a** Immunoblot analysis was performed to assess Pellino1, STAT3, and p-STAT3 (Y705) protein levels in lysates of intestinal macrophages isolated from WT and Pellino1-mKO male mice in normal, acute 1.5% DSS and AOM/DSS groups. Actin was used as a loading control. **b** Immunoblot analysis was performed to assess Pellino1, STAT3, and p-STAT3 (Y705) protein levels in lysates of BMDMs from WT and Pellino1-mKO mice. BMDMs were stimulated with 100 ng/mL LPS for the indicated time. Actin was used as a loading control. **c** GST pulldown assay was performed by incubating RAW cell lysates treated with or without 100 ng/mL LPS for 6 h, along with GST or GST-STAT3. **d** Co-immunization (Co-IP) assays were performed using HEK293T cells transfected with expression vectors for Myc, Myc-tagged Pellino1 (Myc-Pellino1), and Flag-tagged STAT3 (Flag-STAT3). Cellular extracts were immunoprecipitated with an anti-Myc antibody and immunoblotted with an anti-Flag antibody. **e** GST (black arrow) or GST-STAT3 (blue arrow) protein was incubated with purified His-Pellino1 and subjected to immunoblotting using anti-Pellino1 and anti-GST

antibodies. CS: Coomassie brilliant blue staining. **f** WT and Pellino1-mKO BMDMs were stimulated with 100 ng/mL LPS for 6 h. Immunofluorescence staining of Pellino1 (red), p-STAT3 (green), and DAPI (blue). DAPI was used to stain the nuclei. Scale bar = 10 μm. **g** GST pulldown assay was performed using purified GST or GST-Pellino1 protein with RAW cell lysates. RAW cells were stimulated with 100 ng/mL LPS for 6 h. **h** For IP assays, LPS-untreated RAW cells and RAW cells treated with 100 ng/mL LPS for 6 h were lysed. Subsequently, immunoprecipitation was performed using an anti-Pellino1 antibody, followed by immunoblotting with an anti-STAT3 antibody. **i** Co-IP assays were performed using HEK293T cells transfected with expression vectors for Myc-Pellino1, Flag-STAT3 WT, and Flag-STAT3 dominant negative (Flag-STAT3 DN; Flag-STAT3 Y705F). Cellular extracts were subjected to immunoprecipitation using an anti-Myc antibody and subsequently immunoblotted with anti-Flag and anti-Myc antibodies. Source data are provided as a Source Data file.

of p-STAT3 were evident in WT but not in Pellino1-mKO macrophages, and were further increased in LPS-treated WT in accordance with the increased levels of Pellino1. In addition, Pellino1-deleted BMDMs exhibited markedly lower levels of ubiquitinated p-STAT3 in the cytoplasm and nucleus compared to WT BMDMs (Supplementary Fig. 16e). These results indicate that Pellino1 directly regulates the ubiquitination of p-STAT3.

To further specify that Pellino1 regulates the stability of p-STAT3 through K63-linked ubiquitination, we first compared the stability of p-STAT3 between WT and Pellino1-deficient BMDMs treated with cycloheximide, a protein synthesis inhibitor (Fig. 8a). While levels of p-STAT3 in WT BMDMs remained relatively stable, those in Pellino1-deficient BMDMs were sharply reduced. Moreover, overexpression of Pellino1 prolonged the stability of p-STAT3 (Fig. 8b). We also evaluated the involvement of K63-linked ubiquitination in the stability of p-STAT3. Overexpression of K63 increased the half-life of p-STAT3, whereas the K63R mutant failed to do so (Fig. 8c). These data suggest that Pellino1 plays a crucial role in stabilizing p-STAT3 through K63-linked ubiquitination. We subsequently investigated whether Pellino1-mediated STAT3 ubiquitination affects the subcellular distribution of STAT3. In previous results, p-STAT3 was not observed in Pellino1-deficient macrophages, although it was observed in WT macrophages (Fig. 6f). We further isolated cytoplasmic and nuclear fractions of WT and Pellino1-deficient macrophages, respectively. Interestingly, p-STAT3 was observed in both cytoplasmic and nuclear fractions of WT macrophages, whereas it was barely present in the cytoplasmic fraction and not observed in the nuclear fraction of Pellino1-deficient macrophages under conditions where Pellino1 protein was increased by LPS treatment (Fig. 8d).

It is well known that activated STAT3 can induce the expression of target genes by directly binding to promoters of target genes as a transcriptional factor[49]. Therefore, we investigated whether Pellino1-mediated STAT3 ubiquitination affects its binding to the promoter DNA of the target gene as a transcription factor (Fig. 8e). Importantly, our chromatin immunoprecipitation (ChIP) assay revealed a robust binding of STAT3 to the consensus motif in WT BMDMs under LPS-treated condition, which activated STAT3 by phosphorylation. In contrast, the binding of STAT3 to the consensus motif (TTN5AA[50]) was significantly decreased in Pellino1 deficient BMDMs. Consistent with ChIP results, mRNA levels of representative STAT3 target genes, including *MMP9*, *BCL2*, and *IL6*, were reduced in Pellino1-deficient BMDMs compared to WT BMDMs (Fig. 8f). Additionally, we observed significantly lower levels of *MMP9*, *BCL2*, and *IL6* in Pellino1-deficient macrophages compared to those in WT macrophages of both DSS and AOM/DSS-treated groups (Fig. 8g). Notably, mRNA levels of *VEGF*, a key mediator of angiogenesis in cancer, were decreased in Pellino1-deficient macrophages compared to those in WT macrophages in the AOM/DSS-treated group (Fig. 8g). Taken together, these results

suggest that macrophage Pellino1 leads to K63-mediated ubiquitination of STAT3 in response to pathogenic receptor signaling, thereby activating STAT3.

## Discussion

Inflammatory responses within innate immune cells are typically initiated through receptor-mediated signaling[51]. When ligands (cytokines, hormones, and growth factors) bind to receptors, the ubiquitination process can affect the strength and duration of immune responses by modifying PRRs themselves, adaptor proteins, kinases, and other signaling molecules[52]. Ubiquitin-modifying enzymes (UMEs) can precisely target PRRs and downstream components to direct subsequent immune responses, inflammasome activation, and signal transduction[53,54]. UMEs also mediate the degradation of specific proteins, either attenuating inflammation or strengthening immune defense against certain pathogens[53,54]. However, UMEs sometimes amplify signals through ubiquitination, causing excessive immune responses that culminate in chronic inflammation and/or autoimmune disorders[1,53,54]. Therefore, UMEs are important in finely controlling immune responses and maintaining an appropriate response to infection. Previous investigations have elucidated the role of Pellino1 as a UME, primarily facilitating the attachment of ubiquitin chains, notably K48 or K63 linkage, to proteins[17,55]. For example, Pellino1 activates NF-κB signaling by catalyzing K63-linked polyubiquitination of RIP1 in response to TLR3 and TLR4 stimulation, ultimately increasing the production of proinflammatory cytokines[17]. Pellino1 also activates pro-tumor immune responses by targeting K48-linked polyubiquitination of PKCθ in response to T cell receptor signals[18]. Our research found an upregulation of macrophage Pellino1 expression in response to various TLR agonists, particularly LPS. Heightened Pellino1 expression was observed in IBD patients and colitis-induced mice. Individuals with colon adenocarcinoma with elevated *Pellino1* expression exhibited reduced survival rates. Moreover, the incidence of colitis and CAC was significantly reduced in mice with macrophage-deficient Pellino1. Our findings strongly support an association of Pellino1 with aberrant activation of the immune system triggered by receptor-mediated signals under inflammatory conditions.

Results of this study clearly demonstrate that Pellino1 plays a key role in inflammation and inflammation-related cancer development by regulating the ubiquitination of STAT3. However, the incidence of cancer was not significantly different between AOM-treated WT and Pellino1-mKO mice, although the incidence of cancer was significantly reduced in AOM/DSS-treated Pellino1-mKO mice compared to that in WT mice. It has been reported that Pellino1 can promote the spontaneous development of various types of cancer in transgenic mice that overexpress Pellino1[18,21,56]. In the present study, the reduction in colon cancer incidence observed in AOM/DSS-treated Pellino1-mKO mice was also attributed to an inhibition of Pellino1. Furthermore, the

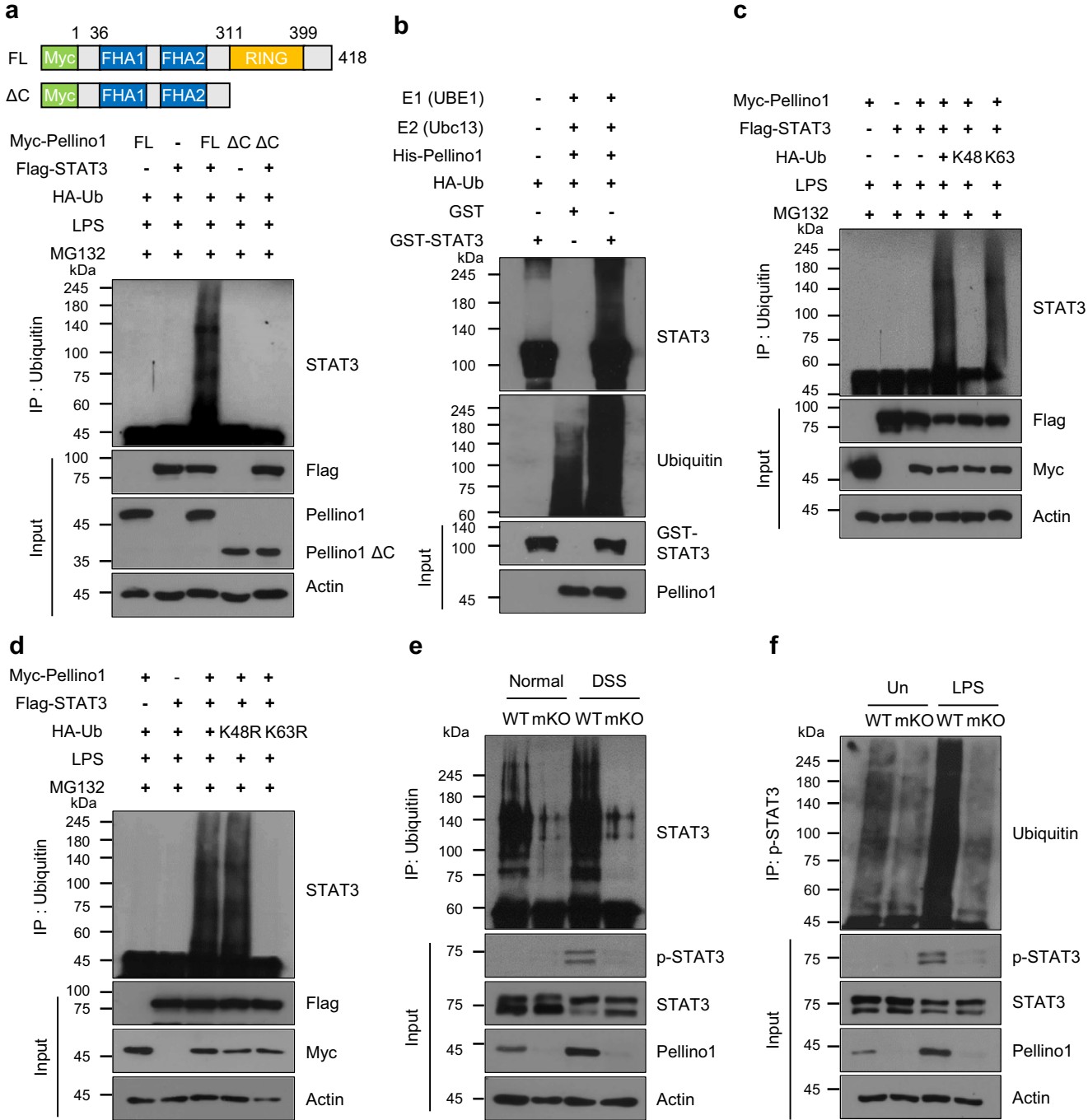

**Fig. 7 | Pellino1 mediates lysine 63-linked ubiquitination of STAT3.**
**a** HEK293T cells were co-transfected with Myc-tagged Pellino1 WT (full-length; FL) or Myc-tagged Pellino1 with a C-terminal deletion (ΔC), Flag-STAT3, and HA-Ub expression plasmids. At 48 h after transfection, cells were treated with MG132 for 5 h, followed by an additional treatment with 1 μg/mL LPS for 30 min. Immunoprecipitation was performed using an anti-Ub antibody. Samples were analyzed by immunoblotting with an anti-STAT3 antibody. The graphic representation depicts Myc-Pellino1 FL and Pellino1 truncation mutants (ΔC), where FHA represents the FHA domain and RING signifies the RING-like domain. **b** In vitro ubiquitination assay of GST-STAT3 was performed using recombinant E1 (UBE1), recombinant E2 (Ubc13), HA-Ub, and His-Pellino1. Mixtures were incubated at 37 °C in an assay buffer containing ATP for 2 h. Immunoblot analysis was performed using anti-STAT3 and anti-Ub antibodies. **c** HEK293T cells were co-transfected with Myc-

Pellino1, Flag-STAT3, along with HA-Ub WT, HA-K48 Ub, and HA-K63 Ub. At 48 h post-transfection, cells were treated with MG132 for 5 h and subsequently stimulated with 1 μg/mL LPS for 30 min. Cell lysates were immunoprecipitated with an anti-Ub antibody and immunoblotted with an anti-STAT3 antibody. **d** HEK293T cells were co-transfected with Myc-Pellino1, Flag-STAT3, along with HA-Ub WT, HA-K48R Ub, and HA-K63R Ub. Experimental procedures were identical to those described in (**c**). **e** Intestinal macrophages were isolated from WT and Pellino1-mKO male mice of normal and acute 1.5% DSS groups. Cells were lysed and immunoprecipitated using an anti-Ub antibody. Immunoprecipitated samples were then analyzed by immunoblotting with an anti-STAT3 antibody. **f** WT and Pellino1-mKO BMDMs were treated with 100 ng/mL LPS for 3 h and then lysed. Cell lysates were immunoprecipitated with an anti-p-STAT3 antibody and immunoblotted with an anti-Ub antibody. Source data are provided as a Source Data file.

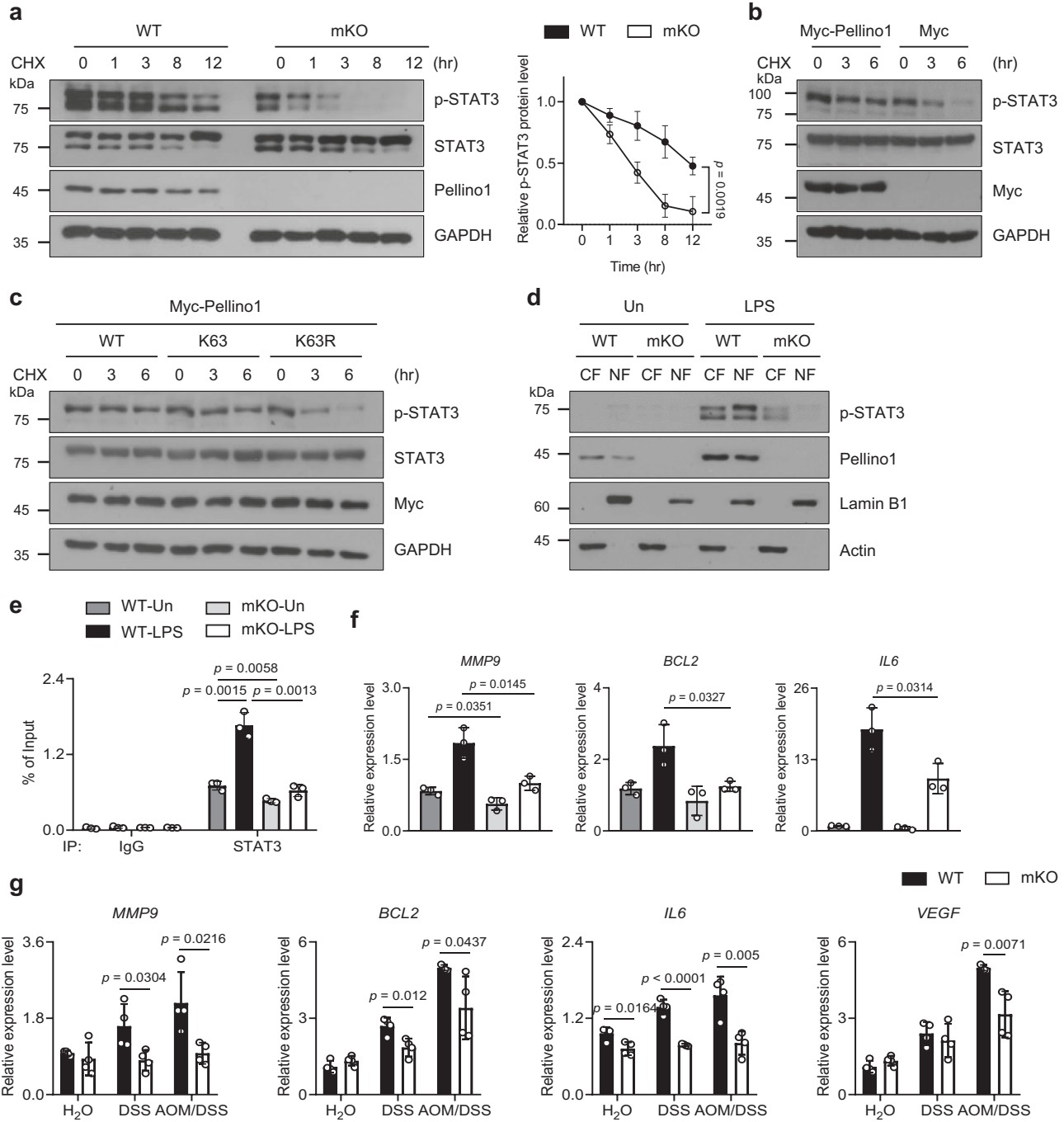

**Fig. 8 | Pellino1-mediated ubiquitination enhances STAT3 activity by increasing the stability of p-STAT3. a** (Left) Immunoblotting of STAT3 and p-STAT3 (Y705) in WT and Pellino1-mKO BMDMs treated with 12.5 μg/mL CHX for the time period indicated in the figure. Cells were stimulated with 100 ng/mL LPS for 3 h before CHX treatment. (Right) Levels of phosphorylated STAT3 were quantified by densitometric analysis of immunoblots using ImageJ (*n* = 4). **b** HEK293T cells were transfected with Myc or Myc-Pellino1. After 48 h of co-transfection, cells were treated with 100 μg/mL CHX for the time period indicated in the figure. Cells were stimulated with 1 μg/mL LPS for 30 min before CHX treatment. **c** HEK293T cells were co-transfected with Myc-Pellino1, HA-Ub WT, HA-K63 Ub, and HA-K63R Ub. Experimental procedures were identical to those described in (**b**). **d** WT and Pellino1-mKO BMDMs were fractionated after 3 h of treatment with 100 ng/mL LPS. Immunoblot analysis was performed to assess p-STAT3 (Y705) protein levels in the

cytoplasm and nucleus. CF cytosolic fraction, NF nuclear fraction. **e** ChIP-qRT-PCR analysis was performed using anti-STAT3 or control IgG with WT and Pellino1-mKO BMDMs as shown in the figure. BMDMs were stimulated with 100 ng/mL LPS for 6 h. ChIP data are presented as relative enrichment normalized to input in WT or Pellino1-mKO BMDMs (*n* = 3). **f** *MMP9*, *BCL2*, and *IL6* mRNA levels in WT and Pellino1 deficient BMDMs (*n* = 3) were quantified with qRT-PCR and normalized against GAPDH expression. **g** *MMP9*, *BCL2*, *IL6*, and *VEGF* mRNA levels in intestinal macrophages from WT and Pellino1-mKO male mice of normal, acute 1.5% DSS, and AOM/DSS groups (*n* = 4) were quantified with qRT-PCR and normalized against GAPDH expression. Data were represented as mean ± SD in (**a**, **e**–**g**). All statistical comparisons were made using a two-tailed Student's *t*-test. Source data are provided as a Source Data file.

expression of Pellino1 was about 1.79 times higher in colitis and 4.9 times higher in CAC than in a normal group (Fig. 4a). These results suggest that Pellino1 can promote colitis and CAC.

A recent study has demonstrated that inhibiting excessive accumulation of macrophages within the immune microenvironment is an attractive strategy for preventing tumor growth and alleviating metastatic diseases[57]. Macrophage recruitment is generally caused by signaling molecules, such as cytokine and chemokine ligands released by immune cells, epithelial cells, stromal cells, and others[58]. This process is regulated by interactions between signaling molecules and receptors on macrophages. However, rapid increases in concentrations of signaling molecules involved in macrophage migration can lead to excessive accumulation of macrophages and persistent inflammation[59]. Therefore, it is important to properly activate macrophages and effectively regulate the accumulation of macrophages through receptor-mediated signaling. A deeper understanding of macrophage behavior has been acquired by studying E3 ubiquitin ligases[60–64]. E3 ubiquitin ligases are renowned for their role in catalyzing ubiquitination, which attaches ubiquitin proteins to target substrates[53,65]. These enzymes play a crucial part in regulating various functions related to macrophage activation, thereby contributing to the development of colitis. For example, myeloid-specific FBXW7 can facilitate EZH2 degradation, upregulate expression of CCL2/7 in macrophages, and enhance the accumulation of CX3CR1[int] proinflammatory macrophages[61]. In addition, TRIM26 can catalyze K11-linked polyubiquitination of TAB1 and promote the expression of proinflammatory cytokines in macrophages through NF-κB and MAPK signaling pathways[62]. Here, we investigated the regulatory role of Pellino1 in macrophage function, particularly in colitis and CAC. Specifically, monocyte-specific ablation of Pellino1 sharply reduced macrophage recruitment and proinflammatory cytokines, thereby attenuating intestinal inflammation during the development of colitis and CAC. Pellino1 in myeloid cells regulates the intrinsic mobility of macrophages. Our investigation also identified a pivotal role of Pellino1 in modulating macrophage phagocytosis during both inflammation and cancer development. Increases of cytokines and chemokines related to macrophage migration within the intestine, including CCL5, CXCL9, CXCL12, CX3CL1, CSF-1, and TGF β, ultimately induced excessive macrophage accumulation in the inflamed intestine. These results strongly suggest that macrophage Pellino1 is a key regulator of macrophage function and a promising therapeutic target for treating IBD and CAC.

Monocytes can migrate, differentiate into macrophages, and polarize into either M1 or M2 macrophages in response to environmental signals[66]. AOM/DSS-treated Pellino1-mKO mice showed reduced recruitment of M2 macrophages compared to WT mice (Supplementary Fig. 5). Pellino1-deficient macrophages also showed lower expression of M2 markers during IL-4/IL-13-induced polarization, although there was no significant difference in M1 polarization induced by LPS and IFN γ (Supplementary Fig. 6). However, their migration ability was significantly reduced in wound healing and migration assays after treatment with LPS, CCL5, and TGF β (Fig. 5c, d and Supplementary Fig. 11). In this regard, we believe that Pellino1 primarily regulates macrophage migration and recruitment. Thus, the decrease in the population of M2 macrophages in AOM/DSS-treated Pellino1-mKO mice was likely due to impaired macrophage recruitment rather than polarization.

STAT3 plays an important role in various autoimmune disorders, including IBD. Initially identified as an acute-phase response factor, STAT3 is an inducible DNA-binding protein associated with the IL-6-responsive element[67]. Subsequent studies have found that STAT3 is activated in response to a wide variety of cytokines (such as IL-6, IL-10, and IL-11), and growth factors (such as fibroblast growth factor and vascular endothelial growth factor) that bind to their receptors[44]. Recent studies have shown that STAT3 can regulate neutrophil migration by activating G-CSF-induced CXCR2 expression[37] and facilitate the recruitment of tumor-associated macrophages, thus influencing the progression of colon cancer[68]. These findings suggest that STAT3-mediated activation of acquired immune responses contributes to the pathogenesis of colitis and CAC by activating and recruiting pathogenic neutrophils and macrophages. Various regulatory factors are involved in regulating STAT3 activation in macrophages. For example, SOCS3 can bind to JAK kinase and cytokine receptors to inhibit STAT3 activation, thereby suppressing the M1-inflammatory phenotype and inactivating inflammatory responses in macrophages[69]. PTP1B can control STAT3 phosphorylation by dephosphorylating JAK2[70]. Inhibition of PTP1B leads to upregulation of heme oxygenase 1 expression via the STAT3 signaling pathway, ultimately promoting M2 polarization[70]. However, knowledge regarding the regulation of STAT3 through ubiquitination within macrophages is limited. Our findings indicate that Pellino1 can function as an E3 ubiquitin ligase with the capacity to modulate STAT3 activation. Pellino1 can promote K63-linked ubiquitination of STAT3, thereby maintaining its phosphorylation and enhancing its activity. This process contributed to the subsequent production of factors related to macrophage migration and proinflammatory cytokines.

In this paper, we propose that Pellino1 is a new regulator that coordinates pathogenic receptor-mediated signaling, such as signaling through the JAK/STAT3 pathway. The Pellino1-STAT3 axis plays a pivotal role in modulating the pathogenesis of colitis and CAC through the K63-linked ubiquitination of STAT3. Elevated levels of Pellino1 in colitis and CAC can result in an increased production of proinflammatory cytokines, enhanced recruitment of macrophages to the colon, and polarization of intestinal macrophages. Further structural studies are necessary to identify precise binding sites between Pellino1 and STAT3 as well as their subsequent effects on ubiquitination. The high selectivity of Pellino1 for phosphorylated STAT3 at the Y705 residue holds promise in the development of an important therapeutic strategy for IBD and CAC, potentially through interventions that can disrupt binding between Pellino1 and STAT3. These results have important implications for understanding inflammation and inflammation-related cancer through the Pellino1-STAT3 signaling axis and for developing effective therapeutic strategies for colitis and CAC (Fig. 9).

## Methods

All animal experiments were conducted in accordance with Institutional Animal Care and Use Committee guidelines (IACUC 2020-11-03-1, IACUC 2021-11-27-3, IACUC 2022-10-34-1, IACUC 2023-01-08-1, IACUC 2023-10-16-1, and IACUC 2024-03-60-1) of Sungkyunkwan University School of Medicine (SUSM). The SUSM is accredited by the Association for Assessment and Accreditation of Laboratory Animal Care International (AAALAC International) in compliance with the Institute of Laboratory Animal Resources (ILAR) guidelines.

### Mice and reagents

Mice were maintained under controlled temperature (23 °C) and humidity (40–60%) conditions. All experimental and control animals were co-housed and maintained on a 12-h light/12-h dark cycle in a specific pathogen-free animal facility. They had free access to food (normal chow diet; #5053, LabDiet) and water except when mice were used in certain experiments described in this paper. All mice used in the experiments were euthanized using $CO_2$ inhalation until their breathing and heartbeat stopped.

The European Mouse Mutagenesis Consortium (EUCOMM) used homologous recombination in C57BL/6NTac embryonic stem cells to insert a cassette containing the coding sequence of the LacZ gene followed by a promoter-driven neo marker flanked by FRT sites between exon 1 and exon 2 of Pellino1. Furthermore, the neo marker and exon 2 of Pellino1 were flanked by loxP sites. Information

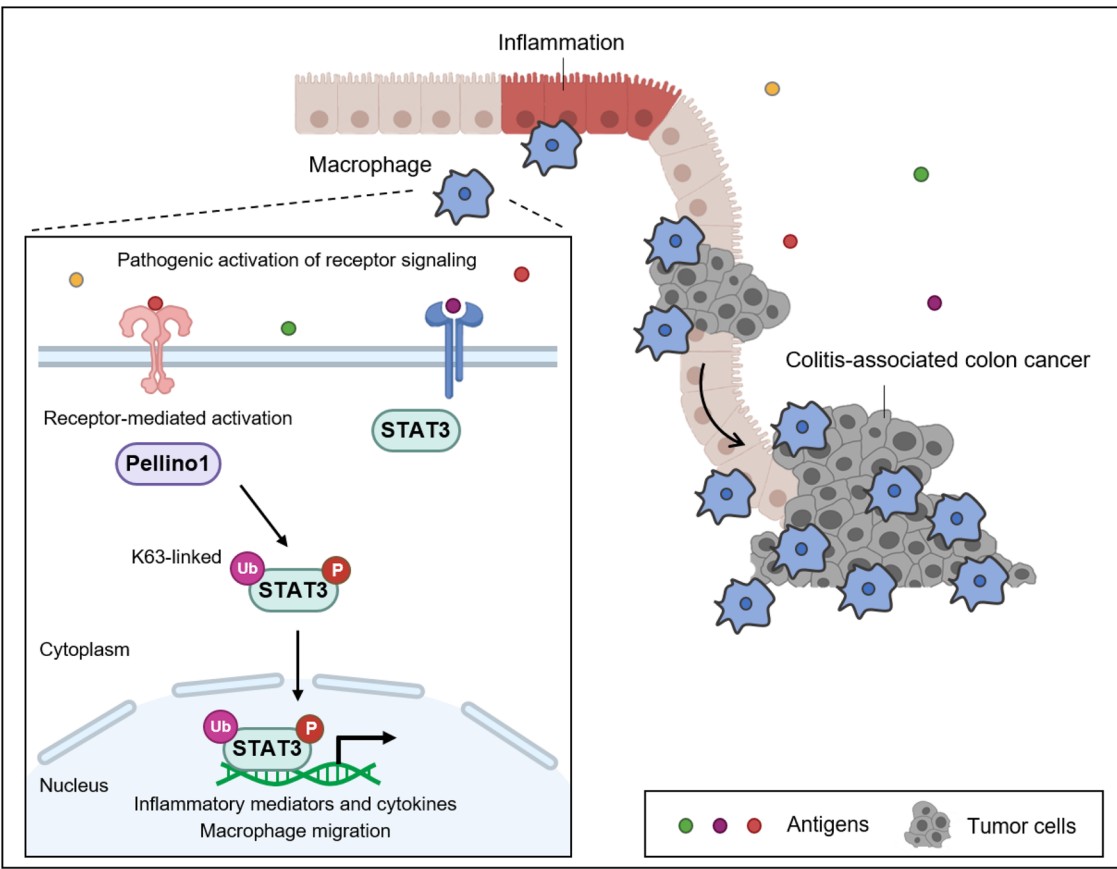

**Fig. 9 | Graphical abstract.** The Pellino1-STAT3 pathway is crucial in the pathogenesis of colitis and CAC through the K63-linked ubiquitination of STAT3.

regarding the Pellino1 knock-out-first reporter-tagged insertion allele (*Pellino1^tm1a*) and the targeting strategy can be found on EUCOMM. To obtain *Pellino1^flox/flox* mice (Pellino1-WT), the *Pellino1^tm1a* allele was crossed with flippase transgenic mice. The Pellino1-WT line was sequentially crossed with *Lysozyme M-Cre* mice to generate mature myeloid-specific Pellino1-knockout (*Pellino1^flox/flox; LysM-Cre*) mice (Pellino1-mKO). Pellino1-mKO mice were generated through a single-generation breeding process. All mice used in this study were based on the C57BL/6 strain. Genotyping was performed by polymerase chain reaction (PCR) analysis. The following primers were used for genotyping: *Pellino1^flox*-F: TGATTGGAAGAGTGAAGCAGCAG, *Pellino1^flox*-R: GTGAACCCAACACCATAATGCTC; *Cre*-F: GTCGATGCAACGAGT-GATGA, *Cre*-R: TCATCAGCTACACCAGAGAC.

Antibodies used in this study were sourced from various suppliers. Antibodies against PCNA (sc-56), MDM2 (sc-812), ERK (sc-94), JNK (sc-7345), GST (sc-138), Pellino1 (sc-271065), GAPDH (sc-25778), Lamin B1 (sc-347015), and ubiquitin (sc-8017) were obtained from Santa Cruz Biotechnology. Antibodies specific for iNOS (13120S), STAT3 (9139S), AKT (4691S), p-AKT (S473) (4060S), p-AKT (T308) (9271S), p-ERK (4376S), p-JNK (4668S), p-p38 (4511S), p-p65 (3033S), p-IKKα/β (2697S), p-STAT3 (Y705) (9145S), p-STAT3 (S727) (9134S), PI3K (4292S), PTEN (9188P), β-Catenin (8480S), HA-Tag (3724S), Myc-Tag (2276S), and K63-linkage-specific polyubiquitin (5621S) were supplied by Cell Signaling Technology. Flag-Tag antibody (F7425) and Actin antibody (A2066) were purchased from Sigma-Aldrich. The Ki67 antibody (GTX16667) was obtained from GeneTex. Antibodies against COX-2 (ab15191), CD68 (ab283654), and Ly6G (Ab25377) were sourced from Abcam. An antibody against F4/80 (MCA497GA) was purchased from Bio-Rad. CD206 antibody (17-2061-82) and NOD2 antibody (14-5858-82) were provided by eBioscience. Reagents used in this study, including Pam3CSK4, HKLM, poly(I:C), ST-FLA, FSL-1, imiquimod, ssRNA40, and ODN1826 CpG DNA, were purchased from Invivogen. M-CSF (416-ML-050) was acquired from R&D Systems. LPS (L5293) and cycloheximide (CHX, C7698) were obtained from Sigma-Aldrich, and MG132 (M-1157) was sourced from AG Scientific. STAT3 inhibitor S3I-201 (CAS 501919-59-1) was purchased from Santa Cruz Biotechnology. NOD2 inhibitor GSK717 (HY-136555) was obtained from MedChemExpress.

## DSS-induced colitis and AOM/DSS-induced colitis-associated colon cancer models

For the DSS-induced colitis model, 8-week-old male mice received a 1.5% DSS (MP Biomedicals) solution in their drinking water (w/v) for a duration of 9 days. These mice were humanely euthanized on the 9th day under anesthesia. Body weight, stool consistency, and the presence of bleeding in each mouse were monitored daily throughout the experiment to calculate the DAI. DAI scores were defined as follows. For weight loss, 0 indicated no loss, 1 represented a 1–5% loss, 2 corresponded to a 6–10% loss, 3 reflected an 11–20% loss, and 4 indicated a loss of more than 20%. For stool consistency, 0 represented normal stool, 2 indicated loose stool, and 4 signified diarrhea. In terms of stool bleeding, 0 represented no blood, 2 indicated slight blood, and 4 reflected gross blood[24]. For experiments confirming Pellino1 expression by IHC staining, 3% DSS in drinking water was administered for 7 days.

For preparing the AOM/DSS-induced colitis-associated colorectal cancer model, 8-week-old male mice were intraperitoneally injected with 10 mg/kg body weight of a colonic carcinogen AOM (Sigma-Aldrich). Subsequently, a 2% DSS (MP Biomedicals) solution was added to drinking water and administered for 7 days, followed by 2 weeks of regular water intake. After completing three cycles of DSS treatment, mice were humanely euthanized under anesthesia on the 84th day. All

experimental procedures were approved by the IACUC. The maximal tumor volume, set at 3000 mm³, was established in accordance with IACUC guidelines, and no animals exceeded this limit during the study. The DSS-induced colitis and AOM/DSS-induced CAC models were created using male mice, as female mice are known to be partially protected from DSS-induced colitis due to the sex hormone estradiol[71].

## Human samples

Endoscopic biopsy specimens of patients with active IBD, including UC and CD, were collected from the pathology database of Asan Medical Center between January and December 2018. The patients were consecutively and randomly collected based on diagnostic criteria. 11 cases of IBD lesions were collected, and 10 cases of normal colon tissue biopsied through health examinations were included as a control group. This retrospective study was approved by the Institutional Review Board (IRB) of Asan Medical Center (IRB No. 2019-0701). The IRB waived the requirement for written informed consent as it was a retrospective study. Colon tissues were subjected to multiplex immunohistochemical (IHC) staining for Pellino1, CD68, and cytokeratin (CK) to investigate expression levels of Pellino1 in inflammatory cells infiltrating colonic lesions of IBD and analyze the relationship between Pellino1-expressing inflammatory cells and macrophages. Tyramide signal amplification (TSA)-based Opal method was used for multiplex IHC staining. Sectioned slides were deparaffinized in xylene, rehydrated in ethanol, washed in distilled water, and fixed in 10% neutral-buffered formalin (NBF) for 10 min. All multiplexed staining procedures were performed with the Opal 7 Immunology Discovery Kit (OP7DS2001KT, Akoya Biosciences). Antigen retrieval (AR) was performed in an AR buffer using microwave treatment (MWT). After MWT, slides were incubated for 15 min and covered with a blocking buffer (ARD1001EA, Akoya Biosciences) for 10 min. Slides were then incubated with primary antibodies against Pellino1 (1:1000, LS-B8877, LS Bio), CD68 (1:700, Opal 7 Immunology Discovery Kit, Akoya Biosciences), CK (1:1000, NCL-L-AE1/AE3, Leica Biosystems), and secondary antibodies (ARH1001EA, Akoya Biosciences) for 1 h. TSA visualization was performed using Opal fluorophores [Opal 520 (Pellino1), Opal 570 (CD68), and Opal 650 (CK)] at 1:150 for 10 min. MWT was further performed to remove the anti-Opal fluorophore complex with AR buffer. All multiplexed staining procedures were performed by repeating MWT through TSA visualization.

A dataset from The Cancer Genome Atlas (TCGA) database of the National Cancer Institute (NCI) was used for data acquisition and analysis of human colorectal adenocarcinoma (COAD) transcriptomes. Data acquisition, processing, and analysis were conducted as described earlier[72]. Transcriptomic data from 448 patients diagnosed with colorectal adenocarcinoma (COAD) were used. The patient cohort included individuals at different stages of the disease, namely stage I ($n = 77$), stage II ($n = 180$), stage III ($n = 127$), and stage IV ($n = 64$). To evaluate survival probability, we utilized the R and its packages, including *multipleROC*, to determine the optimal gene expression value by generating multiple receiver operating characteristic (ROC) curves. *survminer* and *survival* were also employed to determine and visualize the survival probability and Kaplan–Meier curve, as previously reported[73]. Survival analysis was executed by identifying the optimal cutoff point for *Pellino1* gene expression using the Cancer Genome Atlas Colon Adenocarcinoma (TCGA-COAD) dataset (October 2020). This enabled the data to be optimally divided into two distinct groups with the R package, *multipleROC*. The package provided an optimal cutoff value, which was essential for the accurate discrimination of vital status. Using this threshold, samples were then categorized into two groups based on their *Pellino1* gene expression levels. Subsequent survival analysis was performed, and the survival probability was generated by creating Kaplan–Meier curves with the R packages, *survminer* and *survival*. R Studio (ver. 2021.09.1 Build 372,

https://www.rstudio.com/), R (ver. 4.1.2, https://www.r-project. org/), and the R packages "survminer", "survival", "multipleROC", "pROC", "RColorBrewer", "ggpubr", "dplyr", "ggplot2" and "egg" were used for statistical analyses and visualization.

## Cell culture

RAW 264.7 cells (#TIB-71, ATCC), MC38 cells (ENH204-FP, Kerafast), and HEK293T cells (#CRL-3216, ATCC) were cultivated in Dulbecco's modified Eagle medium (DMEM; WELGENE) supplemented with 10% fetal bovine serum (FBS; Gibco) and a combination of antibiotics (WELGENE). These cells were incubated in a humidified incubator at 37 °C with a controlled atmosphere of 5% $CO_2$. BMDMs were generated by culturing bone marrow obtained from femurs and tibiae of 6- to 8-week-old mice in DMEM containing 10% FBS supplemented with M-CSF (20 ng/mL) for a duration of 5 days. The BMDMs were subsequently stimulated with the following reagents: 100 ng/mL Pam3CSK4, $10^7$ cells/mL HKLM, 1 μg/mL poly(I:C), 1 μg/mL LPS, 1 μg/mL ST-FLA, 50 ng/mL FSL-1, 2 μg/mL Imiquimod, 3 μg/mL ssRNA40, and 5 μM ODN1826 for the time periods indicated. For the CHX chase assay, BMDMs were treated with 100 ng/mL LPS for 3 h, followed by treatment with 12.5 μg/mL CHX for 0, 1, 3, 8, and 12 h.

## Isolation of intestinal immune cells

The colon flushed with phosphate-buffered saline (PBS) was longitudinally opened and cut into 1 cm pieces. These colon fragments were then vigorously agitated with 5 mL of 1X Hank's balanced salt solution (HBSS) in 50 mL conical tubes. Subsequently, these tissues were immersed in a Petri dish containing 5 mM EDTA/HBSS for 5 min. This process was repeated five times. Following this, colonic pieces were enzymatically digested in RPMI-1640 medium supplemented with 10% FBS, antibiotics, 1 mg/mL collagenase D (Sigma-Aldrich), and 0.2 mg/mL DNase I (Worthington Biochemical). The digestion was carried out by incubating at 37 °C for 30 min with shaking at 200 rpm. Samples were then passed through 70-μm cell strainers, resuspended in 44% Percoll (Sigma-Aldrich), and layered onto 67% Percoll. Samples were then centrifuged at 800×$g$ for 20 min at room temperature. Cells located at the interface between 67% and 44% Percoll layers were collected and used for subsequent experiments.

## qRT-PCR

Total RNAs were extracted from BMDMs, mouse intestines, and spleens using Qiazol (QIAGEN) or the RNeasy Mini Kit (QIAGEN) following the respective manufacturer's protocols. Complementary DNAs (cDNAs) were synthesized with a cDNA synthesis kit (#G236, Applied Biological Materials Inc. (Abm)) using random primers. Subsequently, cDNAs were used for PCR amplification employing a Power SYBR Green PCR Master Mix (Applied Biosystems) and gene-specific primers. Each sample was processed in triplicate. Relative gene expression levels were determined using the comparative Ct ($2^{-\Delta\Delta Ct}$) method and normalized against GAPDH as a control. All primer sequences are listed in Supplementary Table 1.

## ChIP

Chromatin immunoprecipitation (ChIP) assays were performed using a Simple ChIP Enzymatic Chromatin IP Kit with magnetic beads (9003S, Cell Signaling) according to the manufacturer's instructions. Cross-linked chromatin was digested with micrococcal nuclease, followed by sonication to break it into 150−900 bp fragments. The following antibodies were used: anti-STAT3 (4904S, Cell Signaling), anti-Histone H3 (D2B12) (4620S, Cell Signaling), or Rabbit IgG. For immunoprecipitation reactions, samples were incubated at 4 °C for 4 h with rotation. Enriched fragments were purified and analyzed by qRT-PCR. The signal relative to the input was evaluated using the following formula: percent input = 2% × $2^{(CT\ 2\%\ input\ sample - CT\ IP\ sample)}$, where CT was the threshold cycle of the qRT-PCR reaction. The following qRT-PCR primers were

used: Forward 5′- GATCCTTCTGGGAATTCCTAGATC-3′, Reverse 5′-GATCTAGGACGGCCCAGAAGGATC-3′.

## Flow cytometry

The following fluorochrome-labeled mAbs and staining reagents were used according to the manufacturer's protocol: PE- or PE-Cy7-anti-B220 (clone RA3-6B2, eBioscience), PerCP-Cy5.5- or PE-Cy7-anti-CD3 (clone 145-2C11, eBioscience), FITC- or PE-Cy7-anti-CD4 (clone RM4-5, eBioscience), PE- or APC-anti-CD8 (clone 53-6.7, eBioscience), APC- or PE-Cy7-anti-CD11b (clone M1/70, eBioscience), PE-Cy7-anti-CD11c (clone N418, eBioscience), PerCP-Cy5.5-anti-CD19 (clone 1D3, eBioscience), PerCP-Cy5.5-anti-CD45 (clone 30-F11, eBioscience), PE-Cy7-anti-CX3CR1 (clone SA011F11, BioLegend), FITC-anti-Ly6G (clone 1AB, eBioscience), APC-anti-Ly6C (clone HK1.4, eBioscience), FITC-anti-MHC2 (clone M5/114.15,2, eBioscience), PerCP-Cy5.5-anti-GR-1 (clone RB6-8C5, eBioscience), PE-anti-F4/80 (clone BM8, eBioscience), FITC-anti-CD80 (clone 16-10A1, eBioscience), PE-Cy7-anti-CD86 (clone GL1, eBioscience), APC-anti-CD206 (clone MR6F3, eBioscience), FITC-anti-CD282 (TLR2) (clone 6C2, eBioscience), PE-anti-CD284 (TLR4) (clone HTA125, eBioscience), PE-Cy7-anti-CD369 (Dectin-1) (clone Bg1fpj, eBioscience), and APC-anti-CD36 (clone HM36, eBioscience). Cells were analyzed with a Canto II flow cytometer (BD Biosciences). Flow cytometric analysis was performed using FlowJo software. Flow cytometry results were obtained by gating live cells and excluding cell doublets using FSC/SSC measurements.

## Wound healing assay

A wound healing assay was performed using BMDMs cultured in 24-well plates at a density of $4 \times 10^5$ cells per well. When a confluent cell monolayer was formed, BMDMs underwent a 6 h serum starvation phase. Subsequently, a sterile 100 μL pipette tip was employed to create precise and uniform scratch wounds, followed by a gentle rinse with sterile PBS to eliminate any loosely adherent cells. The culture medium was then replaced with a medium containing a specific stimulant, including 100 ng/mL LPS, 20 ng/mL CCL5, or 100 ng/mL TGF β. Cells were incubated at 37 °C to initiate the wound-healing process. Time-lapse microscopy evaluations were performed at predetermined time intervals (0, 6, 12, and 24 h) to quantitatively assess dynamic cell migration and proliferation. High-resolution images of scratch wounds were acquired using a microscope (Eclipse Ti-U, Nikon) to provide detailed insight into macrophage migration.

## Migration assay

Migration assays were performed using migration chambers with 8 μm pore sizes (Corning). BMDMs at a density of $2.5 \times 10^5$ cells per well were placed in upper chambers of a 24-well plate (Corning) containing 200 μL of serum-free medium. Lower chambers were filled with 750 μL of complete medium diluted with 100 ng/mL LPS, 20 ng/mL CCL5, or 100 ng/mL TGF β. The plate was then incubated at 37 °C overnight. Following incubation, cells were fixed with methanol and stained with 0.1% crystal violet. Cells on the upper surface of the membrane were removed with a cotton swab. Images were captured using an inverted microscope (Eclipse Ti-U, Nikon) and analyzed using NIH ImageJ software.

## Immunohistochemical and immunofluorescence staining

For immunohistochemical staining, colon tissues were fixed in 10% NBF, embedded in paraffin, and sectioned into 5 μm-thick serial sections. These sections were deparaffinized. AR was achieved by heating in 10 mM sodium citrate buffer (pH 6.0). After treatment with a 3% $H_2O_2$ solution, samples were blocked with a buffer containing 3% bovine serum albumin (BSA) and 2.5% normal goat serum. Tissues were incubated with diluted monoclonal antibodies in the blocking buffer and kept refrigerated at 4 °C overnight. The following antibodies were employed: anti-Pellino1 (1:200), anti-F4/80 (1:200), anti-Ki67 (1:200), anti-Ly6G (1:100), anti-PCNA (1:200), anti-β-Catenin (1:200), and anti-MDM2 (1:50). The next day, IHC analysis was performed using a Vectastain Elite ABC kit (Vector Laboratories) in accordance with the manufacturer's instructions. Antibody binding was visualized using 3,3′-diaminobenzidine (DAB). Nuclei were counterstained with hematoxylin (BBC Biochemical). Slides were coverslipped after dehydration. Microscopic images were captured using an AxioCam digital microscope camera (Carl Zeiss) or a confocal laser scanning microscope (LSM710; Carl Zeiss). AxioVision image processing software package (Carl Zeiss) was utilized for image analysis.

A procedure similar to IHC staining was followed for immunofluorescence staining, with the secondary antibody applied on the second day. After treatment with 4′,6-diamidino-2-phenylindole dihydrochloride (DAPI), slides were coverslipped. The following antibodies were used: anti-Pellino1 (1:200), anti-F4/80 (1:200), anti-CD68 (1:100), anti-STAT3 (1:100), anti-p-STAT3 Y705 (1:100), anti-iNOS (1:200), and anti-CD206 (1:200), along with biotinylated anti-rat IgG, anti-mouse IgG, and anti-rabbit IgG secondary antibodies. Microscopic images were captured using an AxioCam digital microscope camera (Carl Zeiss) or a Confocal Laser Scanning microscope (LSM710; Carl Zeiss). For image analysis, the AxioVision image processing software package (Carl Zeiss) was utilized.

## Phagocytosis assay

BMDMs were stimulated with 100 ng/mL LPS for 6 h. MC38 cells labeled with 1 μM 5,6-carboxyfluorescein diacetate succinimidyl ester (CFSE; Biolegend) were co-cultured with BMDMs for 2 h. Subsequently, cells were stained with anti-F4/80 and subjected to flow cytometric analysis. Phagocytic index was calculated using the formula: phagocytic index (%) = (number of F4/80⁺CSFE⁺ cells)/(number of F4/80⁺ cells) × 100.

## Plasmid construction and transfection

Full-length cDNA sequence of the human Pellino1 protein was obtained through PCR amplification using oligo-dT primers. Subsequently, Pellino1 full-length (FL) and Pellino1 ΔC (1–275) variants were subcloned into Myc-tagged plasmids, respectively[21]. Pellino1 ΔC contained 280 amino acids at the N-terminal but lacked the RING domain at the C-terminal. STAT3-Flag plasmid and STAT3 DN-Flag plasmid were procured from Addgene, while HA-tagged Ub (HA-Ub), HA-tagged K48R (HA-K48R), and HA-tagged K63R (HA-K63R) were provided by Ki-Young Lee (Sungkyunkwan University, South Korea). HA-tagged K48 (HA-K48) and HA-tagged K63 (HA-K63) were obtained from Hong Tae Kim (UNIST). All constructs were confirmed by DNA sequencing. For transient transfection, cells were electroporated using a microporator (Digital Biotechnology) following the manufacturer's guidelines. Chemicals and reagents used included 25 μM MG132 and 100 ng/mL LPS.

## Recombinant protein purification

For GST-tagged protein purification, pGEX-KG (GST), pGEX-KG-Pellino1 (GST-Pellino1), and pGEX-KG-STAT3 (GST-STAT3) were expressed in BL21 competent cells through overnight induction at 18 °C with 0.5 mM IPTG. Briefly, pelleted bacteria were resuspended with STE buffer (10 mM Tris–HCl, pH 8.0, 150 mM NaCl, and 1 mM EDTA) and lysed with STE buffer containing 100 μg/mL lysozyme, 0.5 mM phenylmethanesulfonyl fluoride (PMSF), and 1X protein inhibitor cocktail (PIC). After sonication, lysates were centrifuged and resulting supernatants were combined with glutathione Sepharose 4B beads, followed by an overnight incubation at 4 °C. Beads were subsequently washed with 1X PBS. Purified proteins were separated by sodium dodecyl–polyacrylamide gel electrophoresis (SDS–PAGE) and analyzed with Coomassie brilliant blue staining.

## In vitro binding assay

GST or GST-STAT3 protein (2 μg) was incubated with His-Pellino1 (2 μg) in binding buffer (20 mM Tris−HCl (pH 7.4), 100 mM NaCl, 0.5% Nonidet P (NP)-40, 1 mM EDTA, and 1 mM dithiothreitol (DTT)) with glutathione beads for 3 h at 4 °C. These beads were then washed three times with a wash buffer (20 mM Tris−HCl (pH 7.4), 150 mM NaCl, 1% NP-40, 1 mM EDTA, and 1 mM DTT). Bound proteins were eluted with an SDS sample buffer and then separated by SDS−PAGE.

## GST-pull down assay

Cells were lysed with a cell lysis buffer (50 mM Tris−HCl (pH 7.4), 150 mM NaCl, 1% Tween 20, 0.2% NP-40, and 10% glycerol) supplemented with 10 mM NaF, 1 mM Na3VO4, and 1X PIC and centrifuged at 15,000×g for 20 min at 4 °C. A 30 μg sample of the supernatant was retained as input and 1000 μg of proteins were rotated with either GST or GST-STAT3 (2 μg) overnight at 4 °C. These proteins were then washed 3 times with lysis buffer and eluted with SDS sample buffer.

## Immunoprecipitation assay and co-immunoprecipitation

Cells were lysed with immunoprecipitation (IP) lysis buffer (50 mM Tris−HCl (pH 7.4), 150 mM NaCl, 1 mM EDTA, 0.25% sodium deoxycholate, 1% NP-40) supplemented with 1 mM DTT, 1 mM PMSF, and 1X PIC and centrifuged at 15,000×g at 4 °C for 30 min. A 30 μg sample of the supernatant was retained as input and 1000 μg of proteins were rotated overnight at 4 °C with protein A/G beads and an anti-Pellino1 antibody (Santa Cruz Biotechnology). Immunoprecipitated proteins were then washed with lysis buffer 3 times and eluted with SDS sample buffer. A procedure similar to the IP assay was performed for co-immunoprecipitation (Co-IP) using transfected cells. In this experiment, an anti-Myc antibody was incubated with proteins at 4 °C overnight.

## In vivo and in vitro ubiquitination assays

For in vivo ubiquitination assays, HEK293T cells were transfected with an expression plasmid encoding Myc, Myc-Pellino1, or Myc-Pellino1 ΔC, Flag-STAT3, and HA-Ub, HA-K48, HA-K63, HA-K48R, or HA-K63R in indicated combinations. At 48 h after transfection, cells were treated with MG132 (10 μM) for 5 h and then treated with LPS (1 μg/mL) for 30 min as indicated before harvesting them into two tubes. One aliquot was used for conventional immunoblotting as described below. The other aliquot was subjected to IP with anti-Flag antibodies. Immunoprecipitated proteins were washed with lysis buffer three times and eluted with SDS sample buffer.

For the in vitro ubiquitination assay, purified GST or GST-STAT3 (1 μg) protein was incubated with purified His-Pellino1 (100 ng) in conjunction with E1 (50 ng UBE1; Boston Biochem), E2 (400 ng UncH13/Uev1a; Boston Biochem), and HA-Ub (2 μg U-100H; Boston Biochem) in a ubiquitin reaction buffer composed of 500 mM Tris−HCl (pH 7.4), 100 mM MgCl2, 2 mM ATP, and 0.4 mM DTT. Reaction mixtures were incubated at 37 °C for 2 h and then eluted with SDS sample buffer.

## Immunoblotting

Mouse tissues and cells were lysed in RIPA buffer (50 mM Tris−HCl, pH 7.4, 150 mM NaCl, 1% Triton X-100, 0.1% SDS, 1 mM EGTA, 1 mM PMSF, 1 mM Na₃VO₄) supplemented with a mixture of protease inhibitors. Nuclear fractions were obtained using a Nuclear Extraction Kit (ab113474, Abcam) according to the manufacturer's instructions. Protein content in the cell lysate was quantified using a protein assay dye (Bio-RAD). Subsequently, proteins (20−30 μg) were loaded and electrophoresed in 6−10% SDS−PAGE gels in an SDS buffer. Proteins on gels were then transferred onto nitrocellulose (NC) membranes (ATTO) in a transfer buffer (250 mM Tris−HCl and 2 M glycine). These membranes were blocked with 5% skim milk (BD) for 30 min at room temperature. They were then incubated with primary antibodies diluted with 5% milk

or 5% BSA and incubated at 4 °C overnight. Afterward, membranes were incubated with HRP-conjugated goat anti-mouse IgG (SA001, GenDEPOT) or goat anti-rabbit IgG (SA002, GenDEPOT) antibodies. Target proteins were detected using ECL solution (AB frontier). Chemiluminescent signals were developed on film (CU-BU new, Agfa) with an automatic X-ray film processor (JP-33, JPI). Uncropped and unprocessed scans are provided in the Source Data file.

## Statistics and reproducibility

Statistical analyses were performed with GraphPad Prism 8.0.1 software (GraphPad Software). Differences were assessed with two-tailed Student's t-test or ANOVA. p-values < 0.05 were regarded as statistically significant for all tests. Data are presented as mean ± SD or SEM. Survival curves were plotted using the Kaplan−Meier method, and data were compared using the log-rank test. All samples that have received the proper procedures with confidence were included for analysis. Animals and cells were randomized before treatments. n values in the figure legends represent the number of biologically independent samples. Western blots were repeated independently three times with similar results, and representative images are displayed. The tissue slide contained five independent samples, and representative images are shown. Immunofluorescence staining was performed on four independent samples, and representative images are shown.

## Reporting summary

Further information on research design is available in the Nature Portfolio Reporting Summary linked to this article.

## Data availability

Transcriptomic data for 448 patients with colorectal adenocarcinoma (COAD) are downloaded from the Cancer Genome Atlas (TCGA) database. Raw data and uncropped blots corresponding to figures and supplementary figures are included in the Source data. Source data are provided with this paper.

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

## Acknowledgements

This work was supported by grants (NRF-2022R1A2B5B03001431 to C.-W.L.) of the National Research Foundation (NRF) funded by the Ministry of Science and ICT (MSIT), Republic of Korea.

## Author contributions

S.H. designed studies, performed experiments, analyzed data, and wrote the manuscript; J.P., S.-Y.K. and S.-Y.L. performed experiments and analyzed data; H.G., D.R. and Y.J. contributed to specific experiments, analyzed data, and wrote part of the manuscript; C.-W.L. designed studies, supervised the overall project, wrote the manuscript, and performed the final manuscript preparation. All authors provided feedback and approved the final manuscript.

## Competing interests

The authors declare no competing interests.
