## [Transparent Peer Review file · Nature Communications]

The ubiquitin ligase Pellino1 targets STAT3 to regulate macrophage-mediated inflammation and tumor development

Corresponding Author: Professor Chang-Woo Lee

Version 0:

Reviewer comments:

Reviewer #1

(Remarks to the Author)

In this study, the authors investigate the role of the ubiquitin ligase Pellino1 in regulating inflammation and tumor development in colitis-associated colon cancer (CAC). Pellino1 levels were found to be elevated in both patients with colitis and CAC, as well as in murine models of these conditions. By using monocyte-specific Pellino1 knock-out mice, the authors found that Pellino1 deletion in monocytes/macrophages attenuated the development of colitis and CAC by inhibiting macrophage migration and activation. Mechanistically, the authors report that Pellino1 interacts with and targets STAT3 for K63-linked ubiquitination, which correlates with the activation of STAT3.

Overall, the study is performed well, the data is of sufficient quality, and the conclusions are generally supported by the data provided.

However, I have one major concern. The authors claim to show K63-linked ubiquitination of STAT3 by Pellino1. However, I do not think they provide any evidence to substantiate that claim. All experiments the authors perform to address this in cells enriches for STAT3 (overexpressed or endogenous) and then blots for ubiquitin and variants of ubiquitin. Experiments like these cannot prove ubiquitination of the protein of interest (POI), here STAT3. These experiments can only show if STAT3 is found in a complex with ubiquitin. If the authors want to claim and show that STAT3 is ubiquitinated, they need to perform the experiments in the other order: i.e. enrich for ubiquitin (tagged or endogenous using e.g. antibodies or tandem ubiquitin-binding entities, TUBEs), and then blot for POI/STAT3. To show modification of STAT3, STAT3 must then form a smear on the immunoblot with protein species showing significantly reduced mobility and high-molecular weight forms, indicating direct posttranslational modification. I do not think the data provided in this manuscript indicates ubiquitination of STAT3. At best, the *in vitro* ubiquitination assay (Fig 7b) shows monoubiquitination of STAT3 (on additional STAT3 band appearing about the unmodified form), but not polyubiquitination to any level comparable to the ubiquitin smears observed in the other panels in fig 7. This indicates to me that Pellino1 may not ubiquitinate STAT3 but rather ubiquitinate another protein, which interacts with and is coenriched with STAT3, thus giving rise to the ubiquitin signal in the coIP experiments. This issue must be addressed experimentally by the authors by performing the experiments in which they enrich/IP on (poly)ubiquitin and then blot for STAT3 under conditions identical or similar to the ones already presented in fig 7. The authors can read more about guidelines for analysis of ubiquitinated substrates by enrichment and immunoblotting in this paper: <https://pubmed.ncbi.nlm.nih.gov/26325464/>

Reviewer #2

(Remarks to the Author)

This is a study of the ubiquitin ligase Pellino1 which is well known to mediate proinflammatory signaling through ubiquitin ligation. Here, through use of macrophage specific conditional knockout mice (mKO) the authors show that Pellino1 signaling is important for DSS induced colitis and that this further impacts the development of tumors. The authors proceed to attempt the identification of the mechanism behind this effect. The research data is well presented.

Since there is less inflammation in the mKO, it might not be that surprising that there are less tumors in the mKO in the DSS/AOM model, which after all requires inflammation for the accelerated development of tumors. As such Pellino1 may not

per se have anything to do with the tumors, but strictly affects the inflammation. That is supported by the observation that WT and mKO develop equal numbers of tumors in response to AOM only. While the impact of Pellino1 on inflammation is important to report, care should be taken in how the data is interpreted and statements made about direct and indirect significance to cancer. What is missing for the link to cancer is AOM combined with an inflammation model that is independent of Pellino1.

Convincing data is presented that Pellino1 affects STAT3 activation through phosphorylation in vivo. However, the in vitro data that this would be through a direct interaction between Pellino1 and STAT3 is somewhat weak and disconnects somewhat from the finding in vivo. It is clear there is an effect upon STAT3 ubiquitination, but if this is related to activation or not is not shown. Overexpression experiments are prone to artefacts. Some additional experiments are missing to address some essential questions that arises. Can ubiquitinated STAT3 be identified in vivo in WT mice? Can ubiquitinated STAT3 be identified in nuclear extracts in vitro? Can it bind to DNA, specifically the STAT3 consensus DNA binding sequence? Is the ubiquitin removed before nuclear entry? If so, why would it be necessary in the first place, i.e., how would the ubiquitin activate STAT3? In the graphical abstract the ubiquitination is only shown on STAT3 in the cytosol, not in the nucleus, so maybe the authors already think the two observations are not related. Maybe the simple explanation of the in vivo data is that TLRs activate IL-6 in a Pellino1 dependent manner and IL-6 is responsible for the STAT3 phosphorylation. This should be tested.

Additional major specific issues:

1. Fig S1: The conclusion that the Pellino1 signal overlap with CD68 is not convincing. In the overlay, white cells are white and red is still red. The panels a-d imply that co-expression was determined by mathematics – that's a fundamentally flawed experimental design. Furthermore, there is no validation to show the Pellino1 antibody is specific.
2. Fig. 1a: It would be reassuring if QPCR analyses of mRNA was added.
3. The breeding strategy for generating the conditional KO mice is unclear and must be explained. It is stated that Pellino1^{flox/flox} mice (Pellino1-WT) were bred with Lysozyme M-Cre mice to generate mature myeloid-specific Pellino1-knockout (Pellino1^{flox/flox};LysM-Cre) mice (Pellino1-mKO'. Was this a single generation breeding? Where is the Pellino1^{flox/flox} strain from (commercial strain or developed in-house)? How was the strain validated? Without these details the study design cannot be evaluated and if published the study cannot be reproduced.
4. Fig. 2g: It appears 3 colons are show for each strain. The individual colons within each group look vastly different. For the enlarged WT, the focus is on the single mouse that appears to be an outlier. The histology difference between the two groups is not convincing to this reviewer. Colon can be challenging because directionality of sections will affect tissue appearance.
5. Method details for 16s ribosomal RNA (rRNA) bacterial load are missing. This affects reproducibility and potentially study design.
6. In the wound assays, why are the wounds at 0h bigger in the untreated control compared to the treated groups when this is supposed to be 0h? Scale bars should be present and visible on all the images (Fig. 5c, Fig. S8 and Fig. S11).
7. Fig. 6a-b and Fig. 7e: Why is there Pellino1 in isolated Pellino1-mKO macrophages?
8. Fig. S11: How does WT+/- S3I-201 compare to mKO + S3I-201? This should be done to make the connection between Pellino1 and STAT3 in that system.
9. Line 295: It is stated "the application of S3I-201 not only decreased Pellino1 expression", however, what Fig. 11a is showing is that S3I-201 prevents LPS induction of Pellino1 expression, which may not be that surprising.
10. Fig. 6 and Fig. S11d. Pulldown experiments should show an enrichment of the binding partner. That is not the case here and therefore the bands observed are more likely non-specific. In fact, one band in Fig. S11d is labelled as non-specific. Why would only that and not the others be non-specific?

Minor issues.

- 1) While # is often informally used to connote number, it is rarely, if ever, used in formal writing.
- 2) The supplement would be more helpful if figure legends are placed immediately after each figure.
- 3) Lines 101-106: The concepts of TLRs activating Pellino1 function is mixed up with activation of Peli1 gene transcription. Please revise text.
- 4) Fig. 3f: Instead of '2mm<', I would write '>2mm'
- 5) Line 234+238: TLR2, TLR4, CD36, and Dectin-1 do not recognize 'antigen'. Antigen is a term specifically for antibodies.

Reviewer #3

(Remarks to the Author)

The manuscript "The ubiquitin ligase Pellino1 targets STAT3 to regulate macrophage-mediated inflammation and tumor development" by Souen Hwang and colleagues explores links between Pellino1 and development of IBD. By using various animal models, the authors demonstrates very convincingly that Pellino1 contributes to the development of colitis, as suggested by Pellino1-flox LysM-cre animals that shows less signs of colitis and colitis-associated colon cancer. Furthermore, the authors provide excellent data strongly suggesting that Pellino1 controls the activity of STAT3 via K63 ubiquitination. This is a strong manuscript, but I believe some revisions are required:

Major:

- Links between NOD2 and STAT3 were suggested in the past (For instance, <https://doi.org/10.1038/s41598-020-77463-7> and <https://doi.org/10.1038/s41586-021-03484-5>). Since NOD2 is strongly associated with IBD, how NOD2 and Pellino-1 impact each other? Exploring these relationships would strengthen the links between Pellino-1 and IBD.
- Regarding the M1/M2 phenotype: Is Pellino-1 impacting macrophage polarization directly or is this due its effect on

recruitment?

Minor:

- Figures 1D-E: Quantification of Pellino1+/F4/80+ and Pellino1/CD68+ would be helpful
- Line 101: Pellino-1 is induced by TLRs, but I am not sure the reference used (23) is correct
- Figure 2H: Please explain the reason why Ly6G was included in the panel (I understand it is explained a bit better in a later figure, but it is worth it mentioning the rationale in 2H, as it is the first time this appears in the manuscript)
- Figure 2i, Figure 3i: As mentioned in the introduction, Pellino-1 influences IL-10. What are the expression levels of IL-10?
- Supplementary figure 3C: Quantitative analysis would strength these data.
- Figure 4B-E: What is considered PELI1 high and low? Please make it clearer in the text and/or figure.
- Figure 6H: There appears to be substantially less Pellino-1 in the input of lanes 1 and 2, which could Influence the conclusions. Please provide new images.

Version 1:

Reviewer comments:

Reviewer #1

(Remarks to the Author)

The authors have performed a number of new experiments to address my concerns regarding STAT3 ubiquitination. The experiments are now performed in such a way that the authors can conclude that Pellino1 mediates the ubiquitination of STAT3. This has strengthened the manuscript and addressed my main concern.

However, I do not understand the data presented in Fig 7b. In the text, the authors claim this to show that Pellino1 directly ubiquitinates STAT3. The experiment is an in vitro ubiquitination assay with E1, E2, Pellino3, and either GST or GST-STAT3 as substrates. In the figure and the legend, however, it is indicated that the readout is an immunoblot stained with anti-ubiquitin antibodies. But that does not make sense if they want to claim STAT3 ubiquitination. And also, the data don't make sense. In lane 2, E1, E2, Pellino3, and ATP are present (with GST). But no ubiquitin smear is observed. Yet, there should be a lot of ubiquitination here. This is an active reaction with E1, E2, and E3 (Pellino1), and ATP present. They should form a lot of Ub chains regardless of the presence of GST or GST-STAT3. So why is the reaction not active (no Ub smear observed) here when it is clearly active in lane 4, in which the only difference is the presence of GST-STAT3 instead of just GST? Did the authors mislabel the figure? Is it actually a STAT3 blot (not a ubiquitin blot) that is shown?

If it is a ubiquitin blot, something went wrong in the experiment and no conclusion can be drawn from it. And if it is a ubiquitin blot, the experiment doesn't really make sense, because the ubiquitin blot would just show that the in vitro reaction is active and would tell nothing about ubiquitination of STAT3. The authors need to clarify what is going on in this experiment and what is actually shows. And if they want to claim that Pellino1 directly ubiquitinates STAT3, then they need to blot for STAT3 such an in vitro experiment.

Reviewer #2

(Remarks to the Author)

No further comments

Reviewer #3

(Remarks to the Author)

The authors have addressed most of my concerns. However, the data provided regarding the links between NOD2/Pellino1/STAT3 are not strong enough to rule out that nod2 is involved, as only expression levels were provided. To better investigate this, I suggest using Nod2 knockout cells, or use antagonists (such as GSK669). If further experimentation isn't possible, the text should be toned down, as nod2 remains a possibility.

Version 2:

Reviewer comments:

Reviewer #1

(Remarks to the Author)

The authors have addressed my concerns.

Reviewer #3

(Remarks to the Author)

The authors addressed my concerns.

Responses to Reviewer #1

However, I have one major concern. The authors claim to show K63-linked ubiquitination of STAT3 by Pellino1. However, I do not think they provide any evidence to substantiate that claim. All experiments the authors perform to address this in cells enriches for STAT3 (overexpressed or endogenous) and then blots for ubiquitin and variants of ubiquitin. Experiments like these cannot prove ubiquitination of the protein of interest (POI), here STAT3. These experiments can only show if STAT3 is found in a complex with ubiquitin. If the authors want to claim and show that STAT3 is ubiquitinated, they need to perform the experiments in the other order: i.e. enrich for ubiquitin (tagged or endogenous using e.g. antibodies or tandem ubiquitin-binding entities, TUBEs), and then blot for POI/STAT3. To show modification of STAT3, STAT3 must then form a smear on the immunoblot with protein species showing significantly reduced mobility and high-molecular weight forms, indicating direct posttranslational modification. I do not think the data provided in this manuscript indicates ubiquitination of STAT3. At best, the in vitro ubiquitination assay (Fig 7b) shows monoubiquitination of STAT3 (on additional STAT3 band appearing about the unmodified form), but not polyubiquitination to any level comparable to the ubiquitin smears observed in the other panels in fig 7....

Response: We appreciate the reviewer's insightful comments. We also thank the reviewer for giving us an opportunity to address concerns regarding our study on K63-linked ubiquitination of STAT3 by Pellino1. Through various experiments such as immunoprecipitation assay and *in vitro* binding assay (Fig. 6), we observed a direct protein-protein interaction between Pellino1 and STAT3, leading us to hypothesize that Pellino1 functions as the E3 ubiquitin ligase for STAT3. To investigate this, we initially conducted experiments by lysing cells overexpressing Pellino1, STAT3, and ubiquitin. We also used BMDMs. We then performed immunoprecipitation with STAT3 and detected ubiquitin. We also performed *in vitro* ubiquitination assays with GST-STAT3 and His-Pellino1 (Fig. 7b). Unlike the single band typically observed for mono-ubiquitination in a Western blot (Haglund K, et al. Proc Natl Acad Sci U S A 99, 12191-12196, 2002), we observed a smeared band in a sample containing both His-Pellino1 and GST-STAT3, suggesting that Pellino1 mediates the polyubiquitination of STAT3. However, upon reviewing your comments, it seems that our data may have been insufficient to demonstrate that Pellino1 mediates the K63-linked ubiquitination of STAT3.

In response to the reviewer's comment, we tested whether ubiquitination STAT3 could be directly mediated by Pellino1 and performed additional experiments. For this, we transfected 293T cells with

Myc-Pellino1, Flag-STAT3, HA-Ub, HA-K48/HA-K63 (containing only K48 or K63, respectively), or mutant K48 Ub (K48 replaced by R; K48R)/mutant K63 Ub (K63 replaced by R; K63R), followed by immunoprecipitation with anti-ubiquitin antibodies and subsequent immunoblotting with anti-STAT3 antibodies (Fig. 7a, c, d). Results provide direct evidence that Pellino1 mediates the K63-linked ubiquitination of STAT3, rather than STAT3 simply being part of a complex with ubiquitinated proteins. To further confirm this, we conducted *in vivo* ubiquitination assays using intestinal macrophages from WT and Pellino1-mKO mice under normal and DSS-treated conditions (Fig. 7e). Similar to *in vitro* experiments, cell extracts were immunoprecipitated with anti-ubiquitin antibodies, followed by immunoblotting with anti-STAT3 antibodies. As shown in Fig. 7e, ubiquitination of STAT3 was clearly observed in WT macrophages. Ubiquitination of STAT3 was slightly stronger in DSS-treated macrophages. However, the ubiquitination of STAT3 was sharply reduced in Pellino1-mKO macrophages compared to that in WT macrophages. These results support that Pellino1 can act as an E3 ubiquitin ligase that directly ubiquitinates STAT3. However, as the reviewer commented, we do not exclude the possibility that Pellino1 can mediate the ubiquitination of STAT3 via another protein complex formation under certain circumstances.

Responses to Reviewer #2

Since there is less inflammation in the mKO, it might not be that surprising that there are less tumors in the mKO in the DSS/AOM model, which after all requires inflammation for the accelerated development of tumors. As such Pellino1 may not per se have anything to do with the tumors, but strictly affects the inflammation. That is supported by the observation that WT and mKO develop equal numbers of tumors in response to AOM only. While the impact of Pellino1 on inflammation is important to report, care should be taken in how the data is interpreted and statements made about direct and indirect significance to cancer. What is missing for the link to cancer is AOM combined with an inflammation model that is independent of Pellino1.

Response: We thank the reviewer for giving us very valid and important comments. As commented by the reviewer, the key point of this manuscript is that Pellino1-mediated inflammation is a major trigger for the subsequent development of colitis-associated colon cancer (CAC). We agreed with the reviewer that, in this manuscript, if inflammation by Pellino1 was not preceded, Pellino1 alone might not be sufficient to cause colon cancer in the AOM model. What is clear is that macrophage Pellino1 can cause chronic inflammation in response to DSS treatment, followed by colon cancer in a condition treated with AOM.

Additionally, most experiments in this manuscript were performed using macrophage-specific Pellino1 conditional KO mice. We and Sun's group (Park J, et al. *Cancer Immunol Res* 10, 327-342, 2022; Ko, C. J. et al. *EMBO J* 40, e104532, 2021; Park HY, et al. *J Clin Invest* 124, 4976-4988, 2014) have already published results of studies on spontaneous development of various types of cancer caused by Pellino1 in transgenic mice overexpressing Pellino1. In addition, since the development of colon cancer does not show any difference between AOM-treated mice with Pellino1 deficiency and WT mice, we believe that the reduction in colon cancer incidence by AOM/DSS treatment is also affected by the state of Pellino1 expression. However, as commented by the reviewer, the correlation between Pellino1-mediated inflammation and colon cancer was further clarified throughout the manuscript (Results, lines 214-216; Discussion, lines 442-452).

Convincing data is presented that Pellino1 affects STAT3 activation through phosphorylation in vivo. However, the in vitro data that this would be through a direct interaction between Pellino1 and STAT3 is somewhat weak and disconnects somewhat from the finding in vivo. It is clear there is an effect upon

STAT3 ubiquitination, but if this is related to activation or not is not shown. Overexpression experiments are prone to artefacts. Some additional experiments are missing to address some essential questions that arises. Can ubiquitinated STAT3 be identified in vivo in WT mice? Can ubiquitinated STAT3 be identified in nuclear extracts in vitro? Can it bind to DNA, specifically the STAT3 consensus DNA binding sequence? Is the ubiquitin removed before nuclear entry? If so, why would it be necessary in the first place, i.e., how would the ubiquitin activate STAT3? In the graphical abstract the ubiquitination is only shown on STAT3 in the cytosol, not in the nucleus, so maybe the authors already think the two observations are not related.....

Response: We thank the reviewer for pointing this out and we agree with the reviewer. Therefore, we have performed a series of in vitro and in vivo ubiquitination experiments. Results are shown in revised Fig. 7, 8a-c, Supplementary Fig. 15c, 15e, and graphical abstract. As shown in Fig. 8a, the level of p-STAT3 in WT mice remained relatively stable, whereas the level of p-STAT3 was sharply reduced in Pellino1-deficient mice. Furthermore, to determine whether ubiquitination of STAT3 by Pellino1 could be observed not only in the cytoplasm of cells but also in the nucleus, we isolated cytoplasmic and nuclear fractions of macrophages from WT mice and Pellino1-mKO mice, respectively. As shown in Supplementary Fig. 15e, p-STAT3 was observed in both cytoplasmic and nuclear fractions under conditions where Pellino1 protein was increased by LPS treatment.

We subsequently investigated whether Pellino1-mediated STAT3 ubiquitination could affect the subcellular distribution of STAT3. In previous results, p-STAT3 was not observed in Pellino1-deficient macrophages, although it was observed in WT macrophages (Fig. 6f). We further isolated cytoplasmic and nuclear fractions of WT and Pellino1-deficient macrophages, respectively. Interestingly, p-STAT3 was observed in both cytoplasmic and nuclear fractions of WT macrophages. However, it was barely present in the cytoplasmic fraction and not observed in the nuclear fraction in Pellino1-deficient macrophages under conditions where Pellino1 protein was increased by LPS treatment (Fig. 8d).

It is well known that activated STAT3 can induce the expression of target genes by directly binding to promoters of target genes as a transcriptional factor (Xiong YJ, et al. *Aging* (Albany NY) 13, 5185-5196, 2021). Thus, we investigated whether Pellino1-mediated STAT3 ubiquitination could affect its binding to the promoter DNA of the target gene as a transcription factor (Fig. 8e). Notably, our chromatin immunoprecipitation assay revealed a robust binding of STAT3 to the consensus motif in WT BMDMs under LPS-treated condition, which activated STAT3 by phosphorylation. In contrast, the binding of STAT3 to the consensus motif was significantly decreased in Pellino1-deficient BMDMs. Taken together, we believe that macrophage Pellino1 can lead to K63-linked ubiquitination of STAT3

in response to pathogenic receptor signaling, thereby activating STAT3.

Additional major specific issues

1. *Fig S1: The conclusion that the Pellino1 signal overlap with CD68 is not convincing. In the overlay, white cells are white and red is still red. The panels a-d imply that co-expression was determined by mathematics – that's a fundamentally flawed experimental design. Furthermore, there is no validation to show the Pellino1 antibody is specific.*

Response: Supplementary Fig. 1 shows the results of immunofluorescence staining to analyze Pellino1 and CD68 expression in intestinal tissues from healthy individuals and IBD patients. Pellino1 is localized in the cytosol and nucleus, while CD68 is mainly found in lysosomes and endosomes, with some fractions present on the cell surface (Human Protein Atlas, <https://www.proteinatlas.org/>). Due to these distinct localization patterns, complete co-localization of Pellino1 and CD68 within individual cells might not be observed. To clarify, we have revised the description (Results, lines 93-95) to: "the population of CD68⁺ cells expressing Pellino1 in the mucosa of the IBD patients was abundant, compared to the healthy control".

Regarding the specificity of the Pellino1 antibody (LS-B8877, LS Bio), it has been validated in previous studies for detecting Pellino1 in various tissues, including those from patients with diffuse large B-cell lymphoma and psoriasis (Park J, et al. *Cancer Immunol Res* 10, 327-342, 2022; Cho H, et al. *Exp Dermatol* 32, 1476-1484, 2023). This validation supports the reliability and specificity of the antibody used in our experiments.

2. *Fig. 1a: It would be reassuring if QPCR analyses of mRNA was added.*

Response: We thank the reviewer for pointing this out and we agree with the reviewer. Therefore, we treated BMDMs from WT mice with various TLR agonists for 6 hours and measured *Pellino1* mRNA levels by QPCR following cell lysis. QPCR results showed that stimulation with most TLR agonists increased *Pellino1* mRNA levels, with particularly notable elevation observed following LPS treatment (Fig. 1b). These findings indicate that pathogenic receptor signaling can lead to an upregulation of Pellino1 at both mRNA and protein levels, thereby underscoring its crucial role in inflammatory responses.

3. *The breeding strategy for generating the conditional KO mice is unclear and must be explained. It is stated that Pellino1flox/flox mice (Pellino1-WT) were bred with Lysozyme M-Cre mice to generate mature myeloid-specific Pellino1-knockout (Pellino1flox/flox;LysM-Cre) mice (Pellino1-mKO'. Was this a single generation breeding? Where is the Pellino1flox/flox strain from (commercial strain or developed in-house)? How was the strain validated? Without these details the study design cannot be evaluated and if published the study cannot be reproduced.*

Response: Pellino1-mKO mice were generated through a single-generation breeding process by crossing Pellino1 flox/flox mice with LysM-Cre mice. The Pellino1 flox/flox strain was created by crossing Pellino1tm1a allele mice with flippase transgenic mice as illustrated in Supplementary Fig. 3a. Furthermore, genotyping and protein expression assays validated these strains. For genotyping, we used PCR analysis to confirm the presence of LysM-Cre and Pellino1 floxed alleles. Pellino1-WT mice exhibited a 350-bp band, while Pellino1-mKO mice exhibited both 350-bp and 700-bp bands (Supplementary Fig. 3b). For protein expression analysis, we performed Western blot to confirm Pellino1 expression in macrophages from WT and Pellino1-mKO mice (Fig. 2c). We have included additional details in the Materials and Methods of the revised manuscript (lines 586-595).

4. *Fig. 2g: It appears 3 colons are show for each strain. The individual colons within each group look vastly different. For the enlarged WT, the focus is on the single mouse that appears to be an outlier. The histology difference between the two groups is not convincing to this reviewer. Colon can be challenging because directionality of sections will affect tissue appearance.*

Response: The original Fig. 2g showed 1 colon per strain, not 3 individual colons per strain. The method of preparing colons for H&E staining, as depicted in Fig. 3g, might have potentially confused the presence of three distinct colons. To address this, we have revised Fig. 2g to include the whole colon section from two different mice per strain.

5. *Method details for 16s ribosomal RNA (rRNA) bacterial load are missing. This affects reproducibility and potentially study design.*

Response: We investigated epithelial barrier integrity using qRT-PCR with universal 16S primers on spleens from WT and Pellino1-mKO mice in DSS and AOM/DSS groups. We have added these missing details in the qRT-PCR section of the Materials and Methods (lines 666-668).

6. *In the wound assays, why are the wounds at 0h bigger in the untreated control compared to the treated groups when this is supposed to be 0h? Scale bars should be present and visible on all the images (Fig. 5c, Fig. S8 and Fig. S11).*

Response: The wound was created by gently scraping the cell monolayer with a sterile pipette tip. It seemed that we failed to produce a consistent wound. However, the primary objective of this wound-healing experiment was to analyze the movement ability of WT and mKO macrophages. There was no significant difference in wound size between untreated WT and mKO groups at 0 hours. Additionally, we computed the migration as a proportion of the wound area to enable a more precise comparison of macrophage movement. This method could standardize data, facilitating a more accurate evaluation of comparative migration of macrophages between WT and mKO. To enhance the accuracy of our results, we repeated the wound healing analysis and replaced previous data with new data. Moreover, a scale bar was incorporated into all images for improved visualization (Fig. 5c, Supplementary Figs. 10 and 14).

7. *Fig. 6a-b and Fig. 7e: Why is there Pellino1 in isolated Pellino1-mKO macrophages?*

Response: The detection of Pellino1 in Pellino1-mKO macrophages observed in Fig. 6a-b and Fig. 7e (now Supplementary Fig. 15d) is due to incomplete gene deletion. Specifically, experiments in Fig. 6b and Fig. 7e (now Supplementary Fig. 15d) were conducted using BMDMs from WT and Pellino1-mKO mice, while Fig. 6a involved intestinal macrophages isolated from these mice. As shown in Supplementary Fig. 3c-e, there was no significant difference in macrophage population between WT and Pellino1-mKO mice. Equal numbers of macrophages from both genotypes were used in all experiments. Intestinal macrophages were isolated using cell sorting techniques via flow cytometry (Fig. R1). However, due to an incomplete gene deletion inherent to the LysM-Cre model, Pellino1 was still detected, with a deletion efficiency of approximately 79% in BMDMs and 83-98% in mature macrophages (Clausen, B. E. et al. *Transgenic Res* 8, 256-277, 2018). This incomplete deletion resulted in the detection of residual Pellino1 protein.

Fig. R1 (not shown in the manuscript): a, Representative sorting strategy to isolate F4/80⁺CD11b⁺ macrophages. b, Population hierarchy.

8. Fig. S11: How does WT[±] S3I-201 compare to mKO + S3I-201? This should be done to make the connection between Pellino1 and STAT3 in that system.

Response: In response to this comment, we have compared the effects of S3I-201 treatment on macrophage motility in WT and mKO BMDMs. In WT BMDMs untreated with S3I-201, macrophage migration was highly active under stimulation with LPS, CCL5, and TGF β. However, S3I-201 treatment in WT BMDMs significantly inhibited macrophage movement, even in the presence of these stimuli. Treatment with S3I-201 in mKO BMDMs also hardly induced macrophage migration, even when LPS, CCL5, and TGF β were used for stimulation. These findings provide evidence for important functions of Pellino1 and STAT3 in macrophage migration. Additional data have been included in Supplementary Fig. 14c and 14d, along with corresponding descriptions on page 15 (lines 312-318).

9. Line 295: It is stated “the application of S3I-201 not only decreased Pellino1 expression”, however, what Fig. 11a is showing is that S3I-201 prevents LPS induction of Pellino1 expression, which may not be that surprising.

Response: We apologize for our misleading description. As shown in Supplementary Fig. 11a (now Supplementary Fig. 14a), Pellino1 levels in WT BMDMs did not change in response to S3I-201 treatment after 1 hour of LPS stimulation. However, when Pellino1 expression was induced at 3 hours after LPS treatment, S3I-201 treatment slightly reduced Pellino1 levels. To clarify the expression, we have revised lines 313-315 as follows: “S3I-201 did not affect Pellino1 expression in WT BMDMs after 1 hour of LPS stimulation, although it slightly decreased LPS-induced expression of Pellino1 after 3 hours (Supplementary Fig. 14a).”

10. *Fig. 6 and Fig. S11d. Pulldown experiments should show an enrichment of the binding partner. That is not the case here and therefore the bands observed are more likely non-specific. In fact, one band in Fig. S11d is labelled as non-specific. Why would only that and not the others be non-specific?*

Response: We conducted the experiment using a 3.5% input in the GST pull-down assay, which might lead to a less pronounced interaction between Pellino1 and STAT3 compared to other studies that utilized 1% input under similar experimental conditions (Blueggel, M. et al. *Nat Commun* 14, 3258, 2023; Kim, S. H. et al. *Experimental & Molecular Medicine* 52, 1537–1549, 2020). However, when referencing studies that used 3% or more input, the binding observed in our results indicates sufficient interaction (Park HY, et al. *J Clin Invest* 124, 4976–4988, 2014; Lee, J. K. et al. *Cell Mol Immunol* 18, 1395–1411, 2021). Furthermore, the band observed in the original Fig. S11d was labeled as non-specific based on similar findings reported by Park HY et al. (*J Clin Invest* 124, 4976–4988, 2014). They used the same Pellino1 antibody, identified the band at similar positions, also classified it as non-specific. Thus, the conclusion that the band is nonspecific reflects findings of previous studies.

We conducted an additional GST pulldown assay using 1% input with WT BMDMs and observed significant interactions between Pellino1 and STAT3 (Supplementary Fig. 15a), confirming the validity of our findings. These new data were obtained under the same experimental conditions as the original Fig. S11d except for the input amount. To avoid redundancy and present clearest results, we have replaced the old data with the new data (Supplementary Fig. 15a).

Minor issues

1) *While # is often informally used to connotate number, it is rarely, if ever, used in formal writing.*

Response: We thank the reviewer for pointing this out and we agree with the reviewer. Therefore, we have revised the text by replacing the "#" symbol with the appropriate formal expression for positive cell numbers to enhance the clarity and formality of the manuscript.

2) *The supplement would be more helpful if figure legends are placed immediately after each figure.*

Response: We thank the reviewer for pointing this out and we agree with the reviewer. Therefore, we have revised the supplementary information by placing each figure legend immediately after the corresponding figure.

3) Lines 101-106: *The concepts of TLRs activating Pellin1 function is mixed up with activation of Peli1 gene transcription. Please revise text.*

Response: We thank the reviewer for pointing this out and we agree with the reviewer. Therefore, we have revised the manuscript following the reviewer's comments and suggestions (lines 100-105).

4) Fig. 3f: Instead of '2mm<', I would write '>2mm'

Response: We thank the reviewer for pointing this out and we agree with the reviewer. Therefore, we have corrected the text as suggested (Fig. 3f).

5) Line 234+238: *TLR2, TLR4, CD36, and Dectin-1 do not recognize 'antigen'. Antigen is a term specifically for antibodies.*

Response: We thank the reviewer for pointing this out and we agree with the reviewer. Therefore, we have revised lines 246 and 251, replacing 'antigen' with more appropriate terms.

Responses to Reviewer #3

Links between NOD2 and STAT3 were suggested in the past (For instance, <https://doi.org/10.1038/s41598-020-77463-7> and <https://doi.org/10.1038/s41586-021-03484-5>). Since NOD2 is strongly associated with IBD, how NOD2 and Pellino-1 impact each other? Exploring these relationships would strengthen the links between Pellino-1 and IBD.

Response: According to recent studies (SA Gurses, et al. Scientific Reports 10, 20519, 2020; Udden, S. M. N., et al. Cell Rep 19, 2756-2770, 2017), NOD2 deficiency is associated with increased activation of STAT3. We then investigated whether Pellino1's regulation of p-STAT3 was related to NOD2. For this, we analyzed NOD2 expression in intestinal macrophages isolated from WT and Pellino1-mKO mice after DSS treatment. As shown in Supplementary Fig. 13a, there was no significant difference in NOD2 expression between WT and Pellino1-mKO macrophages. However, Pellino1-deficient macrophages exhibited a significantly reduced level of p-STAT3 Y705 compared to WT macrophages after DSS treatment.

Furthermore, LPS treatment resulted in increased NOD2 levels in both WT and Pellino1-deficient BMDMs. However, the extent of such an increase of NOD2 was similar between the two genotypes (Supplementary Fig. 13b). Nonetheless, we observed a positive correlation between p-STAT3 Y705 and Pellino1 expression in LPS-treated BMDMs, with WT BMDMs exhibiting higher levels of p-STAT3 Y705 than Pellino1-deficient BMDMs. These results support that Pellino1 does not regulate STAT3 activation through modulation of NOD2 expression. We have added these statements to appropriate paragraphs in the revised manuscript (Results, lines 305-310).

Regarding the M1/M2 phenotype: Is Pellino-1 impacting macrophage polarization directly or is this due its effect on recruitment?

Response: We thank the reviewer for this comment. To investigate whether Pellino1 directly affected macrophage polarization, we treated WT and Pellino1-deficient macrophages with specific cytokines: LPS and IFN γ to induce M1 macrophages, and IL4 and IL13 to induce M2 macrophages. We then evaluated the expression levels of M1 and M2 marker genes. As shown in Supplementary Fig. 6a, there were no significant differences in the expression of M1 markers (*NOS2*, *CD86*, *TNFA*) between WT and Pellino1 mKO. However, gene expression levels of M2 markers (*ARG1*, *CD163*, *IL10*) were significantly lower in Pellino1 mKO than in WT (Supplementary Fig. 6b), indicating that Pellino1 could

influence M2 macrophage polarization.

When we evaluated the migratory capacity of WT and Pellino1-deficient macrophages under stimulation by LPS, CCL5, and TGF β (Fig. 5c, d and Supplementary Fig. 10), we observed that Pellino1 mKO macrophages had a significantly reduced migratory capacity compared to WT macrophages. Given that LPS can induce M1 macrophages (Zhang C, et al. Front Immunol 12, 620510, 2021), our findings suggest that Pellino1 is more important for regulating macrophage migration than for regulating polarization. Furthermore, it is known that monocytes can migrate, differentiate into macrophages, and then become polarized (Chaintreuil P, et al. Front Immunol 14, 1178337, 2023). Therefore, the reduced recruitment of M2 macrophages observed in Pellino1-deficient mice under CAC conditions (Supplementary Fig. 5) was primarily due to impaired macrophage recruitment rather than a direct effect on macrophage polarization. We have added these statements to appropriate paragraphs in the revised manuscript (Results, lines 186-193; Discussion, lines 482-492).

Minor

1. *Figures 1D-E: Quantification of Pellino1⁺/F4/80⁺ and Pellino1/CD68⁺ would be helpful*

Response: Following the reviewer's comment, we have added graphs quantifying numbers of Pellino1⁺F4/80⁺ and Pellino1⁺CD68⁺ cells (Fig. 1e, Supplementary Fig. 2).

2. *Line 101: Pellino-1 is induced by TLRs, but I am not sure the reference used (23) is correct*

Response: We apologize for the incorrect reference in line 101 (now line 100). We have removed the original sentence and replaced it with a new statement that better aligns with the revised content. We have also updated the reference to match the new sentence.

3. *Figure 2H: Please explain the reason why Ly6G was included in the panel (I understand it is explained a bit better in a later figure, but it is worth it mentioning the rationale in 2H, as it is the first time this appears in the manuscript)*

Response: We have added a brief explanation in lines 134-136.

4. *Figure 2i, Figure 3i: As mentioned in the introduction, Pellino-1 influences IL-10. What are the*

expression levels of IL-10?

Response: We appreciate the reviewer's suggestion. To address this question, we have measured IL10 expression levels in the intestines of WT and Pellino1 mKO mice of DSS and AOM/DSS groups (Fig. 2j and Fig. 3j). Pellino1 mKO mice in the DSS group displayed elevated IL10 mRNA levels compared to WT mice. However, in the AOM/DSS group, Pellino1-mKO mice exhibited lower IL10 levels than WT mice. These results could be due to IL10's role in maintaining a balance between pro-inflammatory and anti-inflammatory immune responses (Carlini V, et al. *Front Immunol* 14, 1161067, 2023). IL10 is a major anti-inflammatory cytokine associated with IBD in humans and mice (Ouyang W, et al. *Annu Rev Immunol* 29, 71-109, 2011). However, IL10 facilitates tumor evasion from immune surveillance within the tumor microenvironment (Mannino MH, *Cancer Lett* 367, 103-107, 2015). These results suggest that Pellino1 can influence IL10 expression depending on the pathological context.

5. Supplementary figure 3C: Quantitative analysis would strength these data.

Response: In response to the reviewer's comment, we have performed a quantitative analysis using ImageJ for intestinal tissues of WT and Pellino1 mKO mice in both normal and AOM/DSS-treated groups. We calculated percentages of positive staining areas for PCNA, β -catenin, and MDM2, and included a new graph of these results in Supplementary Fig. 4c to strengthen our data.

6. *Figure 4B-E: What is considered PELI1 high and low? Please make it clearer in the text and/or figure.*

Response: Fig. 4b-e demonstrates the survival probability based on PELI1 gene expression levels categorized into high and low groups in patients with colon adenocarcinoma. To determine thresholds for high and low PELI1 expression, we utilized R and its packages, including multipleROC. This package generated multiple receiver operating characteristic (ROC) curves to identify the optimal cutoff value for PELI1 gene expression. Using this optimal threshold, we divided samples into two distinct groups: PELI1 high and PELI1 low. We have added details to the Method Section of the revised manuscript (Materials and Methods, lines 560-571).

7. *Figure 6H: There appears to be substantially less Pellino-1 in the input of lanes 1 and 2, which could influence the conclusions. Please provide new images.*

Response: We have updated immunoblots in revised Fig. 6g to provide clearer and more consistent results. These revised images now show improved clarity of Pellino1 bands.

Responses to Reviewer #1 :

The authors have performed a number of new experiments to address my concerns regarding STAT3 ubiquitination. The experiments are now performed in such a way that the authors can conclude that Pellino1 mediates the ubiquitination of STAT3. This has strengthened the manuscript and addressed my main concern.

However, I do not understand the data presented in Fig 7b. In the text, the authors claim this to show that Pellino1 directly ubiquitinates STAT3. The experiment is an in vitro ubiquitination assay with E1, E2, Pellino3, and either GST or GST-STAT3 as substrates. In the figure and the legend, however, it is indicated that the readout is an immunoblot stained with anti-ubiquitin antibodies. But that does not make sense if they want to claim STAT3 ubiquitination. And also, the data don't make sense. In lane 2, E1, E2, Pellino3, and ATP are present (with GST). But no ubiquitin smear is observed. Yet, there should be a lot of ubiquitination here. This is an active reaction with E1, E2, and E3 (Pellino1), and ATP present. They should form a lot of Ub chains regardless of the presence of GST or GST-STAT3. So why is the reaction not active (no Ub smear observed) here when it is clearly active in lane 4, in which the only difference is the presence of GST-STAT3 instead of just GST? Did the authors mislabel the figure? Is it actually a STAT3 blot (not a ubiquitin blot) that is shown?.....(partially omitted)..... If it is a ubiquitin blot, something went wrong in the experiment and no conclusion can be drawn from it. And if it is a ubiquitin blot, the experiment doesn't really make sense, because the ubiquitin blot would just show that the in vitro reaction is active and would tell nothing about ubiquitination of STAT3. The authors need to clarify what is going on in this experiment and what is actually shows. And if they want to claim that Pellino1 directly ubiquitinates STAT3, then they need to blot for STAT3 such an in vitro experiment.

We express our gratitude to the reviewer for the assessment of the manuscript together with the constructive comments. We agree that there are confounding results related to the data presented in Fig. 7b, regarding the direct ubiquitination of STAT3 by Pellino1, and apologize for the confusion. We have edited the manuscript in lines 1201-1202 to better fit the rationale of the study, as well as clarified in the results section (lines 355-356): "We also observed that purified Pellino1 directly catalyzed STAT3 ubiquitination in vitro (Fig. 7b)."

The direct ubiquitination of STAT3 by Pellino1 was studied by isolating the intestinal macrophages and bone marrow-derived macrophages from WT and Pellino1-mKO mice. Next, anti-STAT3 was immunoprecipitated and immunoblotted with anti-Ubiquitin

(Fig. 7f) and vice versa (Fig. 7e), showing how Pellino1 directly affects the ubiquitination of STAT3.

In our initial results, the ubiquitin band was present, but we selected a shorter exposure time for a clearer and stronger signal in line 4 (GST-STAT3). We have incorporated the reviewer's suggestion and have now added the in vitro ubiquitination assay results with STAT3, using recombinant E1, E2, and Pellino1 (revised Fig. 7b). The Pellino1-mediated ubiquitination of STAT3 was then confirmed through immunoblotting with both anti-STAT3 and anti-ubiquitin antibodies (revised Fig. 7b). Our revised blot shows a strong ubiquitination reaction of GST-STAT3, and the revised results have been added to the text and figure legends accordingly: (lines 1201-1202) "Immunoblot analysis was performed using anti-STAT3 and anti-Ub antibodies."

Responses to Reviewer #3 :

The authors have addressed most of my concerns. However, the data provided regarding the links between NOD2/Pellino1/STAT3 are not strong enough to rule out that nod2 is involved, as only expression levels were provided. To better investigate this, I suggest using Nod2 knockout cells, or use antagonists (such as GSK669). If further experimentation isn't possible, the text should be toned down, as nod2 remains a possibility.

We thank the reviewer for the comment and have now modified the text (Page 15, lines 309-311): "Treatment with an NOD2 inhibitor also did not affect Pellino1 expression (Supplementary Fig. 13c, d). These results suggest that Pellino1 is unlikely to regulate STAT3 activation via NOD2."

We treated RAW cells and WT bone marrow-derived macrophages with the NOD2 inhibitor, GSK717, followed by LPS treatment. No significant differences were observed in the Pellino1 expression levels between the untreated and NOD2 inhibitor-treated groups (Supplementary Fig. 13c, d). Furthermore, despite the inhibition of NOD2, Pellino1 was still observed to be clearly induced in response to LPS treatment, suggesting that Pellino1 may function independently of NOD2 in the regulation of macrophage-mediated inflammation. However, we do not rule out the potential involvement of NOD2 in Pellino1-mediated STAT3 regulation, which may depend on the type of pathogenic receptor signal.